# Hippocampal oxytocin receptors are necessary for discrimination of social stimuli

Tara Raam[1,2,3,4], Kathleen M. McAvoy[1,2,3], Antoine Besnard[1,2,3], Alexa H. Veenema[5] & Amar Sahay[1,2,3,4,6]

Oxytocin receptor (Oxtr) signaling in neural circuits mediating discrimination of social stimuli and affiliation or avoidance behavior is thought to guide social recognition. Remarkably, the physiological functions of Oxtrs in the hippocampus are not known. Here we demonstrate using genetic and pharmacological approaches that Oxtrs in the anterior dentate gyrus (aDG) and anterior CA2/CA3 (aCA2/CA3) of mice are necessary for discrimination of social, but not non-social, stimuli. Further, Oxtrs in aCA2/CA3 neurons recruit a population-based coding mechanism to mediate social stimuli discrimination. Optogenetic terminal-specific attenuation revealed a critical role for aCA2/CA3 outputs to posterior CA1 for discrimination of social stimuli. In contrast, aCA2/CA3 projections to aCA1 mediate discrimination of non-social stimuli. These studies identify a role for an aDG-CA2/CA3 axis of *Oxtr* expressing cells in discrimination of social stimuli and delineate a pathway relaying social memory computations in the anterior hippocampus to the posterior hippocampus to guide social recognition.

[1] Center for Regenerative Medicine, Massachusetts General Hospital, Boston, MA 02114, USA. [2] Harvard Stem Cell Institute, Cambridge, MA 02138, USA. [3] Department of Psychiatry, Massachusetts General Hospital, Harvard Medical School, Boston, MA 02114, USA. [4] Program in Neuroscience, Harvard Medical School, Boston, MA 02115, USA. [5] Department of Psychology, Michigan State University, East Lansing, MI 48824, USA. [6] Broad Institute of Harvard and MIT, Cambridge, MA 02142, USA. Correspondence and requests for materials should be addressed to A.S. (email: asahay@mgh.harvard.edu)

Social recognition in mammals is a complex biological process that necessitates, in part, communication between neural circuits subserving discrimination of socially relevant stimuli and those mediating expression of affiliation or avoidance. Alterations in circuits underlying social discrimination or social interaction or the pathways that link these circuits may underlie social recognition deficits seen in autism spectrum disorders and other psychiatric disorders[1–3]. One general mechanism by which distinct behaviors such as reward seeking, social exploration, and discrimination of social stimuli are orchestrated is through the actions of neuromodulators such as oxytocin (OT)[4–7]. Although a growing number of studies have begun to shed light on OT receptor (Oxtr) signaling in the nucleus accumbens, piriform cortex, lateral septum (LS), prefrontal cortex, and ventral tegmental area in social interaction[8–17], the role of hippocampal Oxtrs in social memory processing is not known.

Evidence from classical lesion studies and more recently, from optogenetic and genetic manipulations implicates the anterior and posterior hippocampus in social memory processing[18–24]. Genetic silencing of anterior CA2 (aCA2) population of neurons or excitotoxic lesions of aCA2/CA3[19,22,23] impairs social recognition memory, whereas optogenetic activation of hypothalamic inputs onto aCA2 promotes social memory[24]. Basolateral amygdala inputs to the posterior hippocampus modulate social interaction[20] and posterior CA1 (pCA1) via its projections to the nucleus accumbens is both necessary and sufficient to mediate discrimination of social, but not non-social, stimuli[25]. As aCA2/CA3 projects to aCA1, dorsolateral septum (DLS) and pCA1, it is not clear which of these projections relay social memory computations from aCA2/CA3 to brain regions such as the NAcc and prefrontal cortex that subserve social behavior[9–16,26]. Furthermore, because aCA1 is also important for novel object recognition[27,28], it is unclear whether aCA2/CA3 projections to aCA1 mediate discrimination of both social and non-social stimuli.

Radioligand binding studies, knock-in reporter mice and immunohistochemistry studies have begun to illuminate the distribution of Oxtrs in the hippocampus with varying degrees of resolution[24,29–32]. In mice, Oxtr expression has been reported in polymorphic layer of dentate gyrus (DG), and CA2 and CA3, although the precise identity of Oxtr-expressing cell types is poorly characterized. Pioneering studies relying on injection of anti-OT (OT) sera into the posterior hippocampus suggested a role for OT in social exploration[33]. Partial genetic deletion of Oxtr in the forebrain during the postnatal period impaired discrimination of intrastrain, but not inter-strain, social recognition suggesting a role for forebrain Oxtrs in fine discrimination of social stimuli[34]. However, whether this phenotype is due to loss of Oxtr signaling in the hippocampus, LS, ventral pallidum, or other brain regions is not known[34,35].

Studies of contextual discrimination have demonstrated that the aDG-CA3 circuit is critical for resolving interference between similar memories[36,37]. One circuit mechanism utilized to minimize interference between similar memories is population-based encoding of similar inputs in non-overlapping ensembles of neurons. Such a mechanism is thought to support pattern separation, as it facilitates transformation of similar inputs into divergent outputs[38–46]. Although population-based coding in aDG-CA3 has been demonstrated to underlie discrimination of similar contexts or navigation of similar environments[43,44,47–51], it remains unclear whether this mechanism is also used for discrimination of social stimuli. Consistent with this notion, in vivo recordings in CA2 in rats reveal global remapping of place fields in response to presentation of a familiar or novel social stimulus in a context[21]. As Oxtr signaling enhances signal-to-noise processing in both mice and rats[15,52,53], Oxtrs are well

positioned to modulate DG-CA3 encoding functions under conditions in which OT is released (e.g., when processing social information). Based on these observations, we hypothesized that Oxtrs in aDG-CA2/CA3 contribute to discrimination of social stimuli.

Here we demonstrate using genetic and pharmacological approaches that Oxtrs in the aDG-CA2/CA3 circuit are necessary for discrimination of social, but not non-social (object), stimuli. We show using cellular compartment analysis of temporal activity by fluorescence in situ hybridization (catFISH)[54,55] that Oxtrs in aCA2/CA3 neurons recruit a population-based coding mechanism to mediate discrimination of social stimuli. Optogenetic terminal-specific attenuation studies revealed a critical role for aCA2/CA3 outputs to pCA1, but not aCA1, for discrimination of social stimuli. In contrast, aCA2/CA3 projections to aCA1 mediate discrimination of non-social stimuli. Together these studies identify a role for a aDG-CA2/CA3 axis of Oxtr-expressing cells in discrimination of social stimuli and delineate a pathway by which computations underlying social information processing are relayed from anterior hippocampus to the posterior hippocampus, to guide social recognition.

## Results

**Characterization of *Oxtr* expression in the hippocampus.** To comprehensively map the expression of Oxtr in the hippocampus, we performed multiplex FISH using riboprobes directed against Oxtr, glutamate decarboxylase 1 (Gad1), a marker for GABAergic inhibitory neurons, and markers of interneuron subpopulations, Parvalbumin (PV), and Somatostatin (SST) in age-matched male and female mice. Within anterior DG (aDG), Oxtr expression colocalized primarily with Gad1 (84.3% in females and 87.4% in males), indicating the identity of these neurons primarily as inhibitory interneurons (Fig. 1a, b). A small population of Oxtr+ cells in the hilus did not colocalize with Gad1 (Fig. 1a), suggesting that these cells are most likely excitatory mossy cells. Oxtr expression was absent from Gad1– cells in the granule cell layer of the DG, indicating that Oxtr is not expressed in dentate granule neurons. Analysis of Oxtr, PV, and SST expression in aDG showed greater colocalization of Oxtr with SST than with PV (Fig. 1c, d). Further, a large percentage of Oxtr cells in aDG were negative for both PV and SST, indicating the presence of additional subclasses of Oxtr+ interneurons that are not captured in PV- and SST-only expressing populations.

Next we analyzed Oxtr expression within anterior CA subfields as follows: aCA3$_{proximal}$, aCA2/CA3$_{distal}$, and aCA1 (schematic shown in Fig. 1l). Quantification of Oxtr intensity revealed enrichment of Oxtr expression in the aCA2/CA3$_{distal}$ region while Oxtr levels in aCA3$_{proximal}$ and aCA1 were comparable to that of background (Fig. 1e, f, g, k). Colocalization analysis of Oxtr and Gad1 in aCA2/CA3$_{distal}$ subregion revealed low overlap (10.1% in females and 8.9% in males, Fig. 1g, h), suggesting that the vast majority of Oxtr-expressing cells in aCA2/CA3$_{distal}$ are excitatory. Assessment of overlap between Oxtr, PV, and SST in aCA2/CA3$_{distal}$ revealed a larger fraction of PV+Oxtr+ cells than SST+Oxtr+ cells (Fig. 1i, j). We did not detect sexual dimorphism in Oxtr expression within any subregion of the anterior hippocampus.

Within the posterior hippocampus, we observed dense Oxtr expression in pDG (Supplementary Fig. 1a). In contrast to aDG where the majority of Oxtr cells were GABA-ergic interneurons, >50% of Oxtr-expressing cells in pDG were Gad1– putative mossy cells (Supplementary Fig. 1a, b). Oxtr+ dentate granule neurons were not observed in pDG. Similar to aDG, further characterization of Oxtr+ cells revealed a mixed population of

$PV^+$ and $SST^+$ cells, with a larger fraction of $Oxtr+$ cells colocalizing with $SST$ than with $PV$ (Supplementary Fig. 1c, d).

Next we analyzed $Oxtr$ expression within posterior CA subfields (pCA2/3 and the dorsal and ventral segments of pCA1, schematic shown in Fig. 1l). We observed enriched expression of $Oxtr$ in pCA2/3 and pCA1v; however, $Oxtr$

expression was comparable to the background in pCA1d (Supplementary Fig. 1e, i, j, k). Within pCA2/3, $Oxtr$ is predominantly expressed in excitatory neurons of the pyramidal cell layer, with 15–20% overlap between $Oxtr$ and $Gad1$ (Supplementary Fig. 1e, f). Similar to aCA2/CA3$_{distal}$, these cells are a mixed population of $PV$- and $SST$-expressing cells, with a

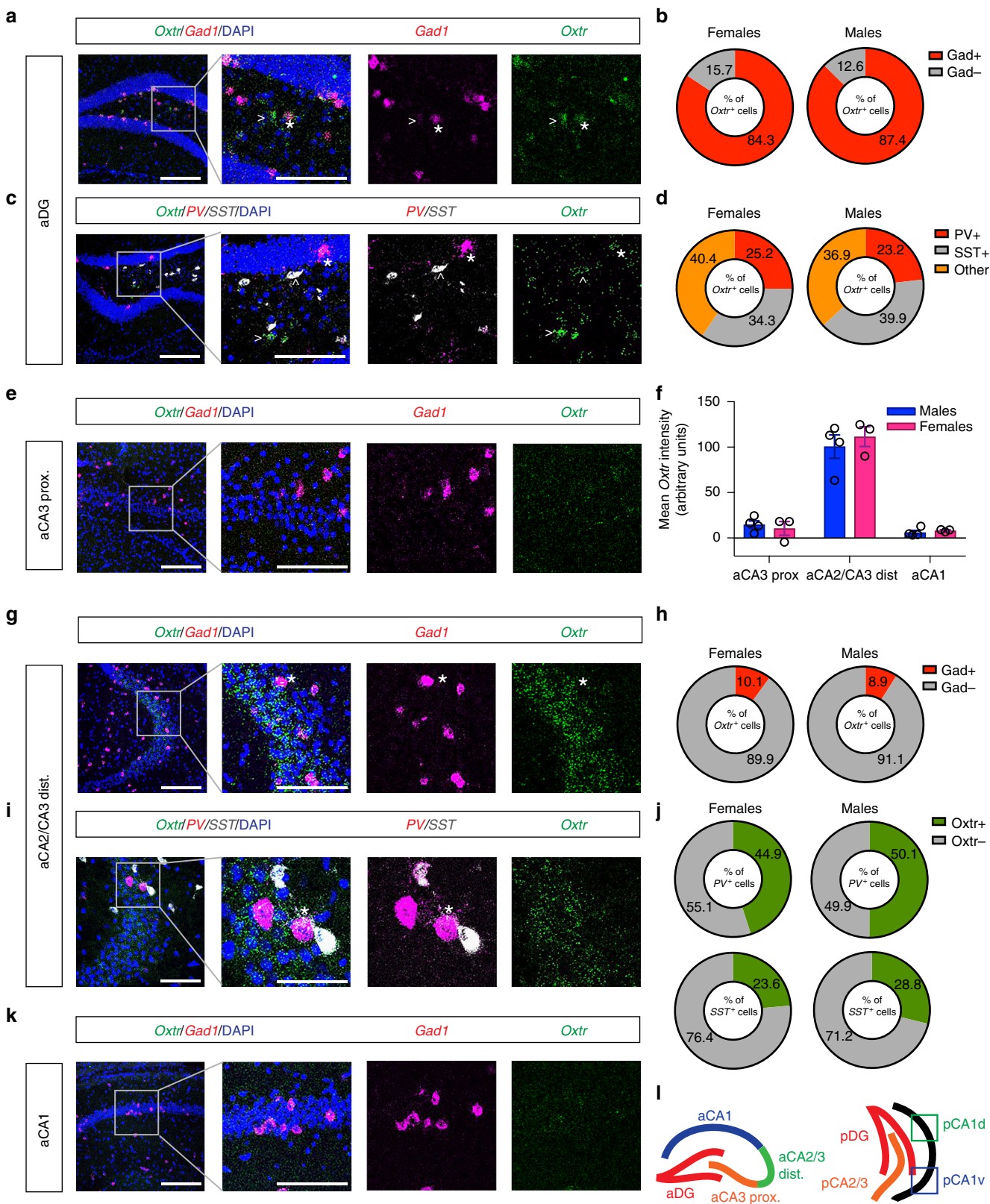

larger fraction of *PV+* cells expressing *Oxtr* than *SST+* cells (Supplementary Fig. 1g, h). In contrast to negligible *Oxtr* expression in aCA1, *Oxtr* expression was dense in the pyramidal layer of the ventral segment of pCA1 (Supplementary Fig. 1k, l). A small fraction of *Oxtr+* cells colocalized with *Gad1*, with a larger fraction of *PV+* cells that were *Oxtr+* than *SST+* cells (Supplementary Fig. 1m,n). We did not detect any sexual dimorphism in *Oxtr* expression within any subregion of posterior hippocampus.

**aDG hilar Oxtrs are necessary for social discrimination**. To determine the behavioral contributions of *Oxtr* in aDG hilar neurons, we utilized a conditional knockout mouse line of the Oxtr, *Oxtr^{f/f}*[35], which permits Cre recombinase-dependent deletion of *Oxtr*. We employed an AAV$_9$ Cre-expressing virus expressed under the control of a human Synapsin promoter, in order to target both inhibitory interneurons and excitatory mossy cells in aDG (Fig. 2a, b). Although previous reports indicate that the AAV$_9$ serotype is anterogradely transported from some cell types, we did not observe any transport to CA3/CA2[56,57] (Supplementary Fig. 2a) or retrograde trafficking to brain regions projecting to the DG (Supplementary Fig. 2b–e), as has also been previously reported with high titre AAV$_9$ viruses[58]. In order to determine viral mediated recombination of *Oxtrs*, we quantified the percentage of DAPI+cells expressing *Oxtr* messenger RNA in aDG hilar neurons of *Oxtr^{+/+}* vs. *Oxtr^{f/f}* mice injected with AAV$_9$-hSyn-Cre. *Oxtr+* cells comprised 11% of all DAPI+ cells in *Oxtr^{+/+}* mice and 6% of all DAPI+ cells in *Oxtr^{f/f}* mice, demonstrative of *Oxtr* recombination (Supplementary Fig. 2f, g).

Bilateral infection of Cre-expressing virus in aDG resulted in an increased number of *c-fos*-expressing dentate granule neurons following exposure to a social stimulus, suggesting a loss of inhibition onto dentate granule neurons (unpaired *t*-test, green fluorescent protein (GFP) vs. Cre: $p = 0.0158$, Fig. 2c). aDG *Oxtr* deletion did not significantly affect performance in tests of object exploration, novel object recognition, or social exploration (Fig. 2d–f). Habituation to a social stimulus was modestly impaired (trend, two-way analysis of variance (ANOVA), interaction: $F_{(2,30)} = 2.613$, $p = 0.0899$). However, aDG *Oxtr* deletion resulted in a robust deficit in the social discrimination task. Whereas control animals strongly preferred a novel animal to a familiar animal, Cre-injected animals did not show a preference (two-way ANOVA, treatment: $F_{(1,15)} = 0.8955$, $p = 0.3590$, stimulus: $F_{(1,15)} = 25.78$, $p = 0.0001$, interaction: $F_{(1,15)} = 10.32$, $p = 0.0058$; Multiple comparisons, Familiar vs. Novel, GFP: $p < 0.0001$, Cre: $p = 0.5294$, Fig. 2g). Discrimination ratios were significantly lower in Cre animals than in GFP

animals (unpaired *t*-test, GFP vs Cre: $p = 0.0036$, Fig. 2g). Unilateral injection of Cre into aDG was insufficient to reproduce this effect (GFP $n = 11$, Cre $n = 7$, unpaired *t*-test GFP vs. Cre: $p = 0.2976$). Deletion of Oxtrs in aDG did not affect measures of innate anxiety in the open field test (OFT) or elevated plus maze (EPM) (Supplementary Fig. 3a–c). Together, these data demonstrate a critical role for Oxtrs in the aDG in mediating discrimination of social, but not non-social, stimuli.

**aCA2/CA3$_{distal}$ Oxtrs are necessary for social discrimination**. As CA2/CA3 is the primary output of the DG and *Oxtrs* are densely expressed in aCA2/CA3$_{distal}$, we next sought to assess the contributions of Oxtr in aCA2/CA3$_{distal}$ to object and social recognition. We preferentially targeted the principal layer of aCA2/CA3$_{distal}$ of *Oxtr^{f/f}* mice by employing a virus that drives expression of Cre-recombinase under a CamKIIα promoter (Fig. 3a, b). Cre virus was expressed predominantly in pyramidal neurons (92.8%) and a small percentage of GABA-ergic interneurons (7.3%) (Supplementary Fig. 5b). We did not observe any retrograde or anterograde transport of Cre virus to inputs or outputs of aCA2/CA3$_{distal}$ in sections immunolabeled for GFP (Supplementary Fig. 4a–c). To determine recombination of *Oxtrs* in aCA2/CA3$_{distal}$, we quantified the percentage of DAPI+ cells expressing *Oxtr* mRNA in aCA2/CA3$_{distal}$ of *Oxtr^{+/+}* vs. *Oxtr^{f/f}* mice injected with AAV$_9$-CamKIIα-Cre. *Oxtr+* cells comprised 50% of all DAPI+ cells in *Oxtr^{+/+}* mice and 25% of all DAPI+ cells in *Oxtr^{f/f}* mice (Supplementary Fig. 4d, e). Interestingly, although we observed fewer total DAPI+ cells that were positive for *Oxtr* in *Oxtr^{f/f}* mice, we observed larger *Oxtr* puncta in cells that retained *Oxtr* mRNA, suggestive of potential redistribution of transcripts.

*Oxtr* deletion in aCA2/CA3$_{distal}$ neurons did not affect object exploration, novel object recognition, or social exploration (Fig. 3c–e). However, although control virus (GFP) injected mice habituated to the familiar animal over trials 1 through 3, Cre-injected animals failed to show this habituation (two-way ANOVA, treatment: $F_{(1,15)} = 0.2574$, $p = 0.6193$, trial: $F_{(2,30)} = 0.6646$, $p = 0.5219$, interaction: $F_{(2,30)} = 6.282$, $p = 0.0053$; Multiple comparisons, Trial 1 vs. Trial 3, GFP: $p < 0.05$, Cre: $p > 0.05$, Fig. 3f). Furthermore, control animals demonstrated a strong preference for the novel animal, whereas Cre-injected animals failed to show a preference (two-way ANOVA, treatment: $F_{(1,15)} = 0.00185$, $p = 0.9663$, stimulus: $F_{(1,15)} = 19.98$, $p = 0.0004$, interaction: $F_{(1,15)} = 5.597$, $p = 0.0319$; Multiple comparisons, Familiar vs. Novel, GFP: $p = 0.0009$, Cre: $p = 0.2439$, Fig. 3f). In addition, discrimination ratios were significantly lower in Cre animals than in GFP animals (unpaired *t*-test, GFP vs. Cre: $p = 0.0038$, Fig. 3f). Deletion of *Oxtrs* in aCA2/CA3$_{distal}$ did not

**Fig. 1** Characterization of *Oxtr* expression in anterior hippocampus. **a** Representative low-magnification (left) and high-magnification (right) images of *Oxtr* and *Gad1* mRNA expression in anterior DG by FISH ($n = 4$ males, 3 females). Asterisk (*) denotes *Oxtr^+*/*Gad1^+* interneuron. Arrowhead (>) denotes *Oxtr^+*/*GAD1^-* mossy cell. Scale bar, 100 μm. Inset scale bar, 50 μm. **b** Quantification of *Gad1* and *Oxtr* colocalization for males and females, expressed as a percentage of *Gad1^+* cells over total *Oxtr^+* cells. **c** Representative images of low-magnification (left) and high-magnification (right) images of *Oxtr*, *PV*, and *SST* mRNA expression in anterior DG by FISH ($n = 4$ males, 3 females). Asterisk (*) denotes *PV^+*/*Oxtr^+* interneuron. Right-facing arrowhead (>) denotes *Oxtr^+* mossy cell. Upward facing arrowhead (^) denotes *Oxtr^+*/*SST^+* interneuron. Scale bar, 100 μm. Inset scale bar, 50 μm. **d** Quantification of *Oxtr* colocalization with *PV* and *SST*, expressed as a percentage of total *Oxtr* cells. **e** Representative low-magnification (left) and high-magnification (right) images of *Oxtr* and *Gad1* mRNA expression in aCA3$_{proximal}$ by FISH ($n = 4$ males, 3 females). Scale bar, 100 μm. Inset scale bar, 50 μm. **f** Quantification of mean *Oxtr* intensity for CA regions expressed in arbitrary units, normalized for background. **g** Representative low-magnification (left) and high-magnification (right) images of *Oxtr* and *Gad1* mRNA expression in aCA2/CA3$_{distal}$ by FISH ($n = 4$ males, 3 females). Asterisk (*) denotes *Oxtr^+*/*GAD1^+* interneuron. Scale bar, 100 μm. Inset scale bar, 50 μm. **h** Quantification of *Gad1* and *Oxtr* colocalization in aCA2/CA3$_{distal}$ for males and females, expressed as a percentage of *Gad1^+* cells over *Oxtr^+* cells. **i** Representative images of *Oxtr*, *PV*, and *SST* mRNA expression in aCA2/CA3$_{distal}$ by FISH ($n = 4$ males, 3 females). Asterisk (*) denotes *PV^+*/*Oxtr^+* interneuron. Scale bar, 50 μm. Inset scale bar, 25 μm. **j** Quantification of *Oxtr* colocalization with *PV* and *SST* in aCA2/CA3$_{distal}$, expressed as a percentage of total *PV* cells (top) or *SST* cells (bottom). **k** Representative low-magnification (left) and high-magnification (right) images of *Oxtr* and *Gad1* mRNA expression in aCA1 by FISH ($n = 4$ males, 3 females). Scale bar, 100 μm. Inset scale bar, 50 μm. **l** Schematic of anterior and posterior hippocampal subregions boundaries

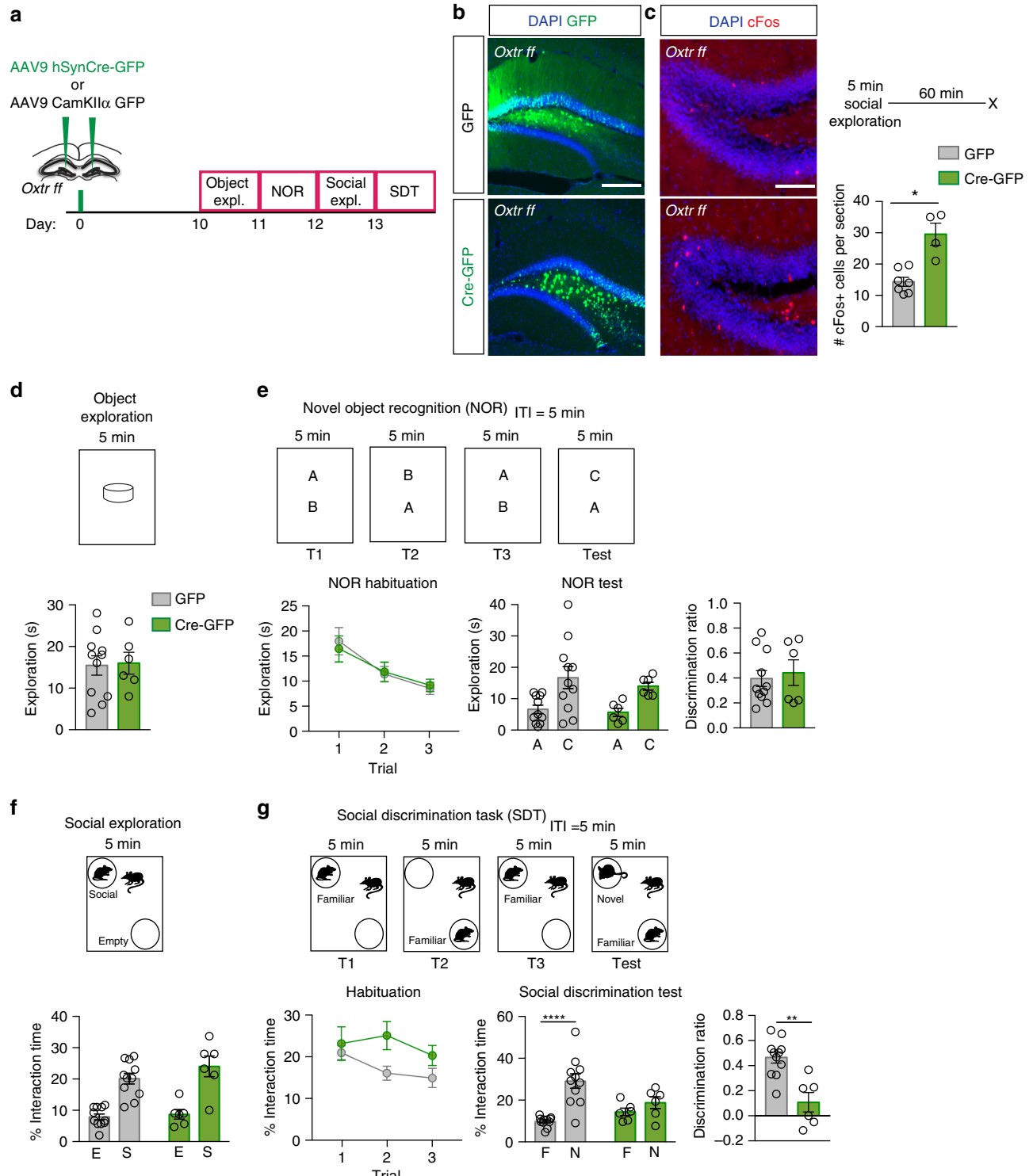

**Fig. 2** Viral recombination of *Oxtr* in anterior DG hilar neurons impairs discrimination of social, but not non-social, stimuli. **a** Schematic illustrating viral injection and behavioral testing timeline. **b** Representative images of Cre and GFP virus infection in aDG. Scale bar, 200 μm. **c** Representative images and quantifications of cFos immunoreactivity in granule cell layer of aDG (GFP: $n = 7$, Cre: $n = 4$). Scale bar, 75 μm. **d** Behavioral schematic (top) and quantification (bottom) of single object exploration (GFP: $n = 11$, Cre: $n = 6$). **e** Behavioral schematic (top) and quantification (bottom) of novel objection recognition (GFP: $n = 11$, Cre: $n = 6$). Quantifications are displayed as Habituation (trials 1–3), Test (trial 4), and discrimination ratio (trial 4). **f** Behavioral schematic (top) and quantification (bottom) of social exploration test (GFP: $n = 11$, Cre: $n = 6$). **g** Behavioral schematic (top) and quantification (bottom) of social discrimination task (GFP: $n = 11$, Cre: $n = 6$). Quantifications are displayed as Habituation (trials 1–3), Test (trial 4), and discrimination ratio (trial 4). All data are displayed as mean ± SEM

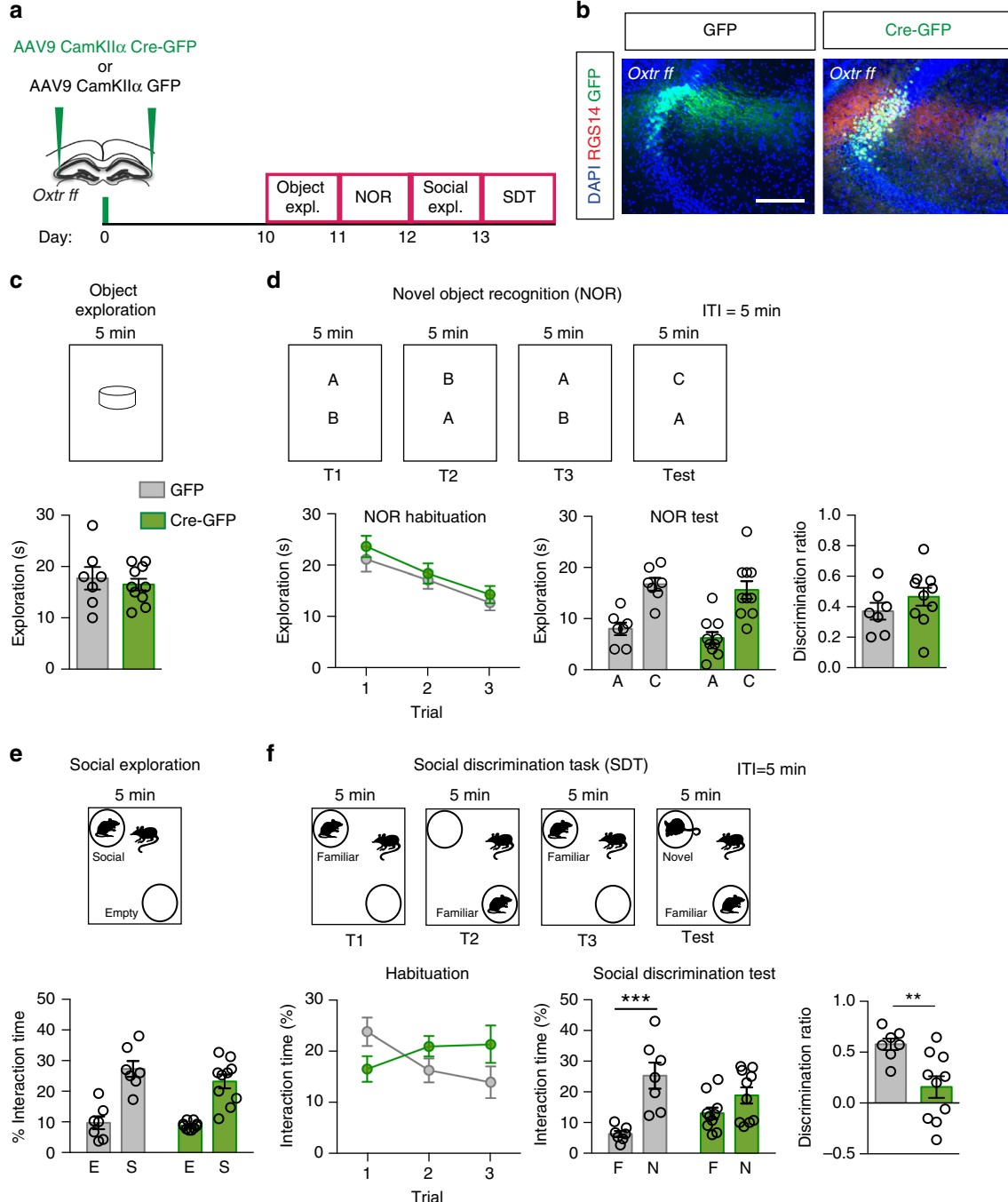

**Fig. 3** Viral recombination of *Oxtr* in aCA2/CA3$_{distal}$ impairs discrimination of social, but not non-social, stimuli. **a** Schematic illustrating viral injection and behavioral testing timeline. **b** Representative images of Cre and GFP virus infection in aCA2/CA3. Scale bar, 200 μm. **c** Behavioral schematic (top) and quantification (bottom) of single object exploration (GFP: $n = 7$, Cre: $n = 10$). **d** Behavioral schematic (top) and quantification (bottom) of novel objection recognition (GFP: $n = 7$, Cre: $n = 10$). Quantifications are displayed as Habituation (trials 1–3), Test (trial 4), and discrimination ratio (trial 4). **e** Behavioral schematic (top) and quantification (bottom) of social exploration test (GFP: $n = 7$, Cre: $n = 10$). **f** Behavioral schematic (top) and quantification (bottom) of social discrimination task (GFP: $n = 7$, Cre: $n = 10$). Quantifications are displayed as Habituation (trials 1–3), Test (trial 4), and discrimination ratio (trial 4). All data are displayed as mean ± SEM

affect measures of innate anxiety in the open field (Supplementary Fig. 5a, c). Interestingly, in the EPM, we observed a decrease in time spent in the center, a decision-making point in the maze, at the expense of time spent in the closed arm, but not time spent in open arms (Supplementary Fig. 5a, d).

To distinguish between the potential contribution of Oxtrs to habituation (or acquisition) vs. retrieval in the social discrimination task, we employed pharmacological blockade of Oxtrs in vivo

only during the retrieval phase of this task. Adult male C57BL/6J mice received a bilateral guide cannulae placed over aCA2/CA3 10 days before behavior (Supplementary Fig. 6a, b). Following habituation and acquisition, we infused sterile saline or Oxtr antagonist, Oxtr-A, (L-368,899 Hydrochloride, Tocris) bilaterally into aCA2/CA3$_{distal}$ after Trial 3 and immediately before Trial 4. Under conditions in which subjects have sufficiently encoded the familiar stimulus, acute blockade of Oxtrs during retrieval was

sufficient to impair discrimination of social stimuli (two-way ANOVA, treatment: $F_{(1,5)} = 2.373$, $p = 0.1841$, stimulus, $F_{(1,5)} = 1.509$, $p = 0.2739$, interaction, group × stimulus: $F_{(1,5)} = 9.802$, $p = 0.0259$; paired $t$-test Familiar vs. Novel, Saline: $p = 0.0148$, Oxtr-A: $p = 0.1672$, Supplementary Fig. 6c). Similarly, discrimination ratios were significantly lower during Oxtr-A infusion than during saline infusion (paired $t$-test, $p = 0.0109$, Supplementary Fig. 6c).

Together, these experiments suggest through genetic deletion that Oxtrs in aCA2/CA3$_{distal}$ are critical for social discrimination, but not object discrimination. Furthermore, our pharmacological blockade studies suggest that Oxtr function is necessary during the retrieval window to mediate discrimination of social stimuli.

### aCA2/CA3$_{distal}$ Oxtrs mediate population-based social coding.

The DG-CA3 circuit recruits population-based coding to encode similar memories in non-overlapping ensembles, thereby minimizing interference between similar memories during storage. Such a mechanism has been shown to underlie navigation and discrimination of similar contexts[43,44,47–51], and has been more recently shown in CA2 of rats in response to social stimuli[21]. We asked whether Oxtrs in aCA2/CA3$_{distal}$ recruit population based coding mechanisms to distinguish between similar social stimuli by utilizing catFISH in combination with viral deletion of *Oxtrs* in aCA2/CA3$_{distal}$. This approach allows us to visualize ensembles of neurons activated in response to social stimuli at distinct temporal windows based on sub-cellular localization of *cFos* mRNA transcripts in either the nucleus or the cytoplasm[54,55] (Fig. 4a, b and Supplementary Fig. 7). Adult male *Oxtr$^{f/f}$* mice were injected with AAV CamKIIα-GFP or CamKIIα-Cre-GFP in aCA2/CA3$_{distal}$ 1 week before exposure to two social stimuli: re-exposure to a single mouse (A–A), or new exposure to a novel mouse (A–B) (four groups in total, Fig. 4c). Analysis of cellular ensembles in CA3$_{proximal}$ of control animals revealed higher percentage overlap (cells positive for both nuclear and cytoplasmic *cFos* transcript are indicative of reactivation) in the A–A condition relative to the A–B condition. In contrast, Cre-injected animals exhibited comparable levels of re-activation (overlap) in A–A and A–B conditions (two-way ANOVA, treatment: $F_{(1,15)} = 0.1703$, $p = 0.6856$, exposure: $F_{(1,15)} = 3.335$, $p = 0.0878$, interaction: $F_{(1,15)} = 5.917$, $p = 0.0280$; Multiple comparisons, AA vs AB, GFP: $p = 0.0145$, Cre: $p > 0.9999$, Fig. 4d). Interestingly, analysis of sub-cellular transcript localization in CA3$_{distal}$ of controls and experimental groups revealed comparable levels of ensemble reactivation in response to A–A and A–B conditions (Fig. 4f), consistent with reports that CA3$_{proximal}$ and CA3$_{distal}$ are functionally distinct[45,46,49]. We did not observe differences in the total number of activated cells for either CA3$_{proximal}$ or CA3$_{distal}$ (Fig. 4e, g and Supplementary Fig.7).

### Roles of aCA2/CA3 distal outputs in social discrimination.

Social information processed in aCA2/CA3$_{distal}$ may be relayed via projections to aCA1, DLS, and/or pCA1 to subcortical and prefrontal circuits, to guide social behavior[59–61] (Supplementary Figs. 8 and 10). To begin to understand how Oxtr-dependent computations underlying social recognition performed in aCA2/CA3$_{distal}$ are relayed out of the anterior hippocampus to limbic circuits that mediate social behavior, we utilized in vivo terminal-specific optogenetic silencing to attenuate aCA2/CA3$_{distal}$ outputs to these three downstream targets. We first targeted the pathway from aCA2/CA3$_{distal}$ to aCA1. We injected AAV$_5$-CamKIIα-eNpHR-EYFP or AAV$_5$-CamKIIα-EYFP bilaterally into aCA2/CA3$_{distal}$ of adult male mice. Three weeks post injection, bilateral optogenetic implants were placed above aCA1. Mice were then tested for both object behavior and social behavior (Fig. 5a–d).

Optogenetic illumination of aCA2/CA3$_{distal}$ terminals in aCA1 in the test phase of novel object recognition task robustly impaired discrimination between familiar and novel object in NpHR animals relative to controls (two-way ANOVA, treatment: $F_{(1,14)} = 3.783$, $p = 0.0721$, object: $F_{(1,14)} = 28.67$, $p = 0.0001$, interaction: $F_{(1,14)} = 15.88$, $p = 0.0014$; Multiple comparisons, Object A vs. C, EYFP: $p < 0.0001$, NpHR: $p = 0.6985$, Fig. 5d). Discrimination ratios of NpHR mice were significantly lower than controls (unpaired $t$-test $p = 0.0029$, Fig. 5d). We did not observe any effects on object exploration, social exploration, or discrimination of social stimuli (Fig. 5c, e, f). These data suggest that aCA2/CA3$_{distal}$ outputs to aCA1 are critical for object recognition but are dispensable for discrimination of social stimuli.

We next asked whether aCA2/CA3$_{distal}$ outputs to the DLS mediate social discrimination[12,14,59,60]. AAV$_5$-CamKIIα-eNpHR-EYFP or AAV$_5$-CamKIIα-EYFP viruses were bilaterally injected into aCA2/CA3$_{distal}$ of adult male mice and bilateral optogenetic implants were placed above the DLS (Supplementary Fig. 9a, b). Optogenetic attenuation of terminals was confirmed by quantifying the number of cFos+cells in the termination area (unpaired $t$-test, $p = 0.0006$, Supplementary Fig. 9c). We observed a modest enhancement in social discrimination (two-way ANOVA, treatment: $F_{(1,30)} = 0.2753$, $p = 0.6037$, stimulus: $F_{(1,30)} = 52.65$, $p < 0.0001$, interaction: $F_{(1,30)} = 4.861$, $p = 0.0353$; Multiple comparisons, Familiar vs. Novel, EYFP: $p = 0.0018$, NpHR: $p < 0.0001$, Supplementary Fig. 9g). Discrimination ratios were significantly higher for NpHR animals than control animals (unpaired $t$-test, $p = 0.0121$, Supplementary Fig. 9g). We did not observe any effects on single object exploration, NOR, or social exploration (Supplementary Fig. 9d, e, f). These data suggest that attenuation of aCA2/CA3$_{distal}$-DLS projections may modestly promote discrimination of social stimuli.

Recent studies have demonstrated a role for pCA1-NAcc circuit in social discrimination[25]. We observed direct outputs from aCA2/CA3$_{distal}$ to the dorsal segment of pCA1 (Fig. 6b)[62] and confirmed the colocalization of these terminals with the CA1 marker, WFS1 (Supplementary Fig. 10). We attenuated these outputs by injecting AAV$_5$-CamKIIα-eNpHR-EYFP or AAV$_5$-CamKIIα-EYFP into bilateral aCA2/CA3$_{distal}$ of adult male mice and placing a bilateral optogenetic implant above pCA1 (Fig. 6a). Effective silencing was confirmed by quantifying the number of cFos+cells in pCA1 below the optogenetic implant (unpaired $t$-test, $p = 0.0223$, Fig. 6c). Optogenetic attenuation of aCA2/CA3$_{distal}$ terminals in pCA1 did not affect performance in tests of single object exploration, novel object recognition, or social exploration (Fig. 6d–f). In contrast, optogenetic illumination of aCA2/CA3$_{distal}$ terminals in pCA1 in the test phase of social discrimination task markedly impaired discrimination in NpHR animals as compared with controls (two-way ANOVA, treatment: $F_{(1,12)} = 0.82$, $p = 0.3830$, stimulus: $F_{(1,12)} = 64.71$, $p < 0.0001$, interaction: $F_{(1,12)} = 13.4$, $p = 0.0033$; Multiple comparison, Familiar vs. Novel, EYFP: $p < 0.0001$, NpHR: $p = 0.0184$, Fig. 6g). Discrimination ratios were also significantly lower for NpHR animals as compared with controls (unpaired $t$-test, $p = 0.0006$, Fig. 6g). Together, these data suggest that aCA2/CA3$_{distal}$ projections to pCA1 relay social information to subcortical and prefrontal circuits to guide social behavior.

## Discussion

Social recognition, the ability to recognize and distinguish between conspecifics is essential to the formation and stability of social interactions across species. As such, it is not surprising that cortical and subcortical circuits must communicate to support efficient social recognition. Remarkably, despite studies

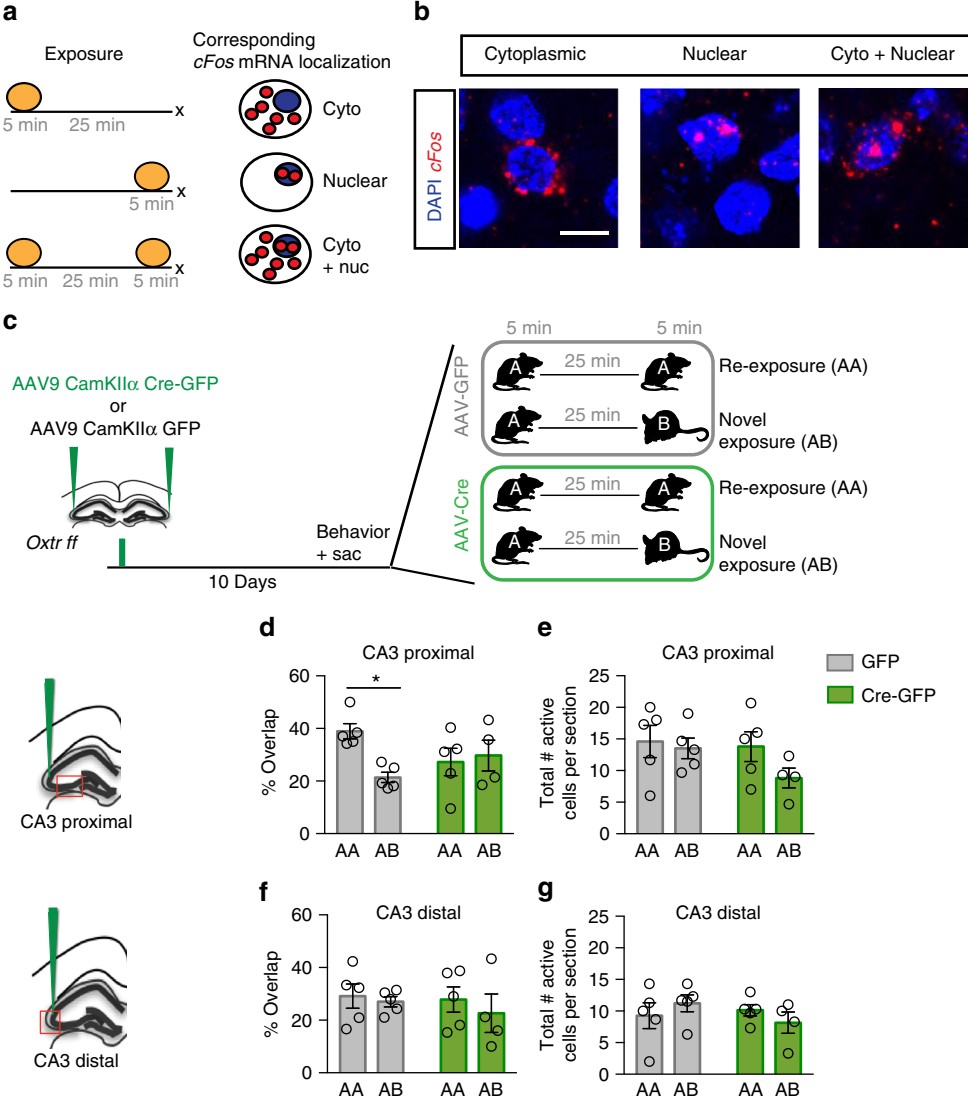

**Fig. 4** Viral recombination of *Oxtr* in aCA2/CA3$_{distal}$ impairs population based coding of social stimuli. **a** Schematic illustrating conceptual design of catFISH using *c-fos* intronic and full-length mRNA localization as readouts for ensembles of neurons activated at distinct behavioral windows. **b** Representative images of CA3 pyramidal neurons exhibiting cytoplasmic, nuclear, or cytoplasmic and nuclear localization of *c-fos* mRNA transcripts. Scale bar, 10 μm. **c** Schematic illustrating viral injection and behavioral exposure timelines (GFP AA: $n = 5$, GFP AB: $n = 5$, Cre AA: $n = 5$, Cre AB: $n = 4$). **d, f** Percent overlap between ensembles of neurons encoding first and second stimulus for CA3$_{proximal}$ (top) and CA3$_{distal}$ (bottom), respectively. **e, g** Total number of active cells per section for CA3$_{proximal}$ (top) and CA3$_{distal}$ (bottom), respectively. All data are displayed as mean ± SEM

implicating Oxtrs in different brain regions in social recognition, the role of Oxtrs in the hippocampus has remained unaddressed. Here we identify a role for a aDG-CA2/CA3 axis of Oxtr-expressing cells in discrimination of social, but not non-social, stimuli. Further, Oxtr deletion in aDG and aCA2/CA3$_{distal}$ did not affect object behaviors or social interaction suggesting a specific role for Oxtrs in fine discrimination of social stimuli, consistent with phenotype of mice lacking Oxtrs in the forebrain[34]. Our findings are consistent with a general role of the aDG-CA3 circuit of the hippocampus in resolving interference between similar memories[37,61,63,64]. The lack of an effect of Oxtr deletion in aDG-CA2/CA3 in object recognition suggests that the same aDG-CA2/CA3 memory circuit scaffold has been co-opted by an ancient evolutionarily conserved neuromodulatory system, OT-Oxtr, to permit encoding when OT is released (such as during encounter of social stimuli)[13].

Recent studies have suggested that CA2 links social information with space or contextual information[21]. However, whether

CA3 exhibits population-based coding in response to similar social stimuli or whether Oxtr engages such population-based coding mechanisms to promote social discrimination is not known. Using catFISH based ensemble imaging we found that the CA3$_{proximal}$, but not CA3$_{distal}$, subregion exhibits population-based coding when mice encountered similar, but not the same, social stimuli. The lack of population-based coding in CA3$_{distal}$ may be explained by changes in firing rate that are not detectable with catFISH. Caveats notwithstanding, these findings are consistent with growing evidence that under conditions of high input similarity (as is the case here with mice of the same genetic background) CA3$_{proximal}$, but not CA3$_{distal}$, performs population-based coding[45,46,49]. Preferential removal of Oxtrs of aCA2/CA3 pyramidal neurons impaired population based coding in CA3$_{proximal}$, suggesting that OT recruits basic circuit mechanisms normally used by CA3$_{proximal}$ to support orthogonalization of similar inputs into divergent outputs (pattern separation) to minimize overlap between social stimuli. These findings are

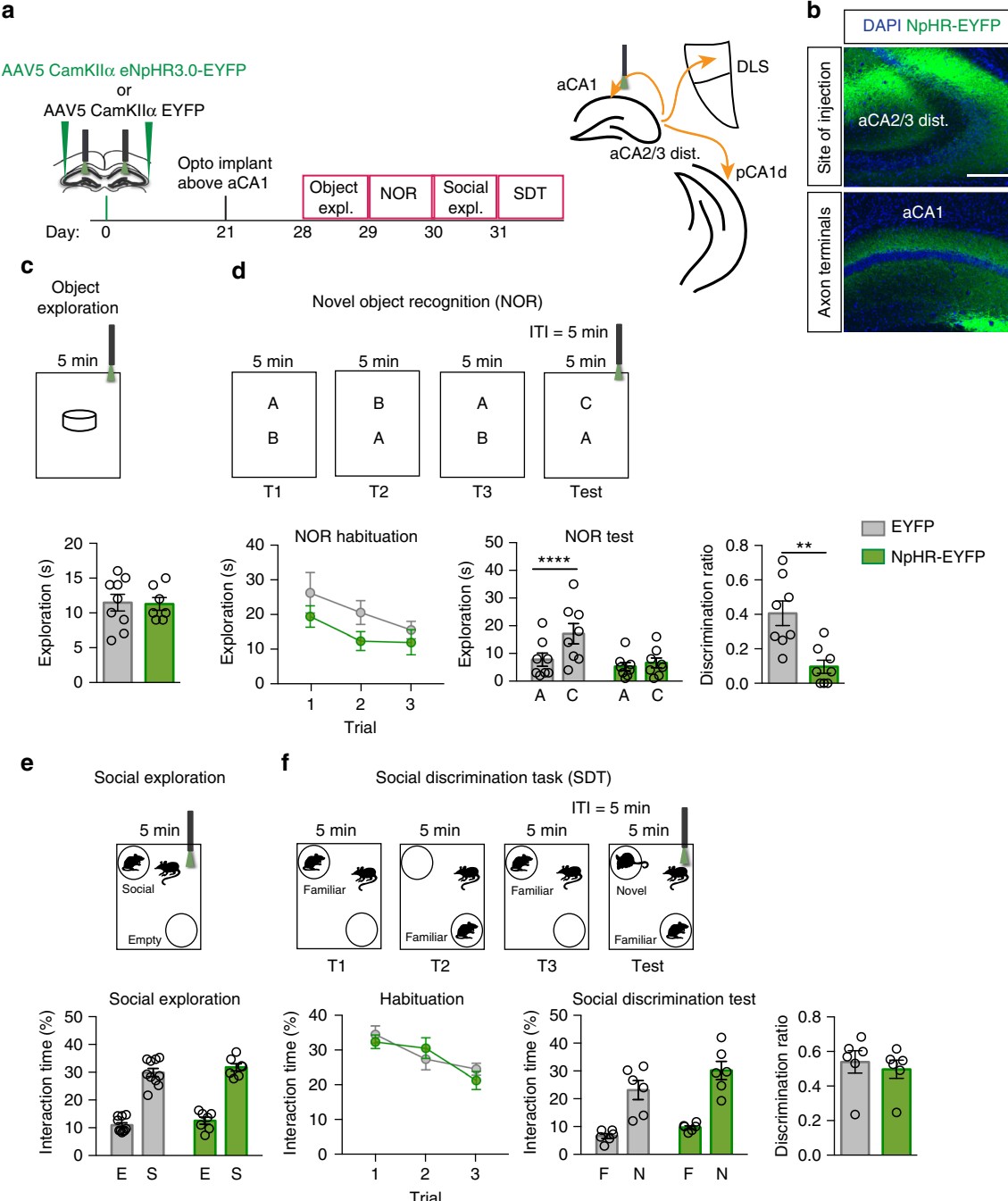

**Fig. 5** Optogenetic attenuation of aCA2/CA3$_{distal}$ outputs to aCA1 impairs discrimination of object, but not social stimuli. **a** Schematic illustrating viral injection, optogenetic implants and behavioral testing timeline. **b** Representative images of site of injection of eNpHR3.0 virus in aCA2/CA3$_{distal}$ cell bodies (top) and corresponding axon terminals in aCA1 (bottom). Scale bar, 200 μm. **c** Behavioral schematic (top) and quantification (bottom) of single object exploration (EYFP: $n = 9$, NpHR: $n = 7$). **d** Behavioral schematic (top) and quantification (bottom) of novel objection recognition (EYFP: $n = 8$, NpHR: $n = 8$). Quantifications are displayed as Habituation (trials 1–3), Test (trial 4), and discrimination ratio (trial 4). Laser was on during trial 4 only. **e** Behavioral schematic (top) and quantification (bottom) of social exploration test (EYFP: $n = 10$, NpHR: $n = 7$). **f** Behavioral schematic (top) and quantification (bottom) of social discrimination task (SDT) (EYFP: $n = 6$, NpHR: $n = 6$). Quantifications are displayed as Habituation (trials 1–3), Test (trial 4), and discrimination ratio (trial 4). Laser was on during trial 4 only. All data are displayed as mean ± SEM

consistent with previous reports that identify functional heterogeneity among CA3 subregions, with CA3$_{proximal}$ behaving as a pattern separator, much similar to the DG, and CA3$_{distal}$, strongly influenced by its auto-associative networks, functioning as a pattern completion network[45,46,49,65]. However, that we see a disruption in population based coding in CA3$_{proximal}$, despite low Oxtr expression in this region, suggests a role for local

microcircuits within CA3 that communicate social information mediated by OT signaling in aCA2/CA3$_{distal}$ to pattern separation networks in CA3$_{proximal}$.

Oxtrs in the auditory and piriform cortices and hippocampus have been shown to enhance signal-to-noise processing through the modulation of excitatory and inhibitory neurons[15,52,53,66]. Interestingly, we observed increased activation of the DG

following recombination of Oxtr in the hilus, suggestive of decreased sparseness, and reminiscent of potentiated social stimulus-induced DG activation in mice lacking OT[67]. Clarification of the functions of Oxtr in aCA2 or aCA3 excitatory neurons and within distinct GABAergic populations (SST, PV,

and other subtypes) and non-GABAergic (presumably mossy cells) cells in the hilus will further edify the contributions of Oxtrs to DG-CA2/CA3 circuit properties.

Our optogenetic terminal-specific attenuation studies revealed a critical role for aCA2/CA3 outputs to pCA1, but not aCA1, for

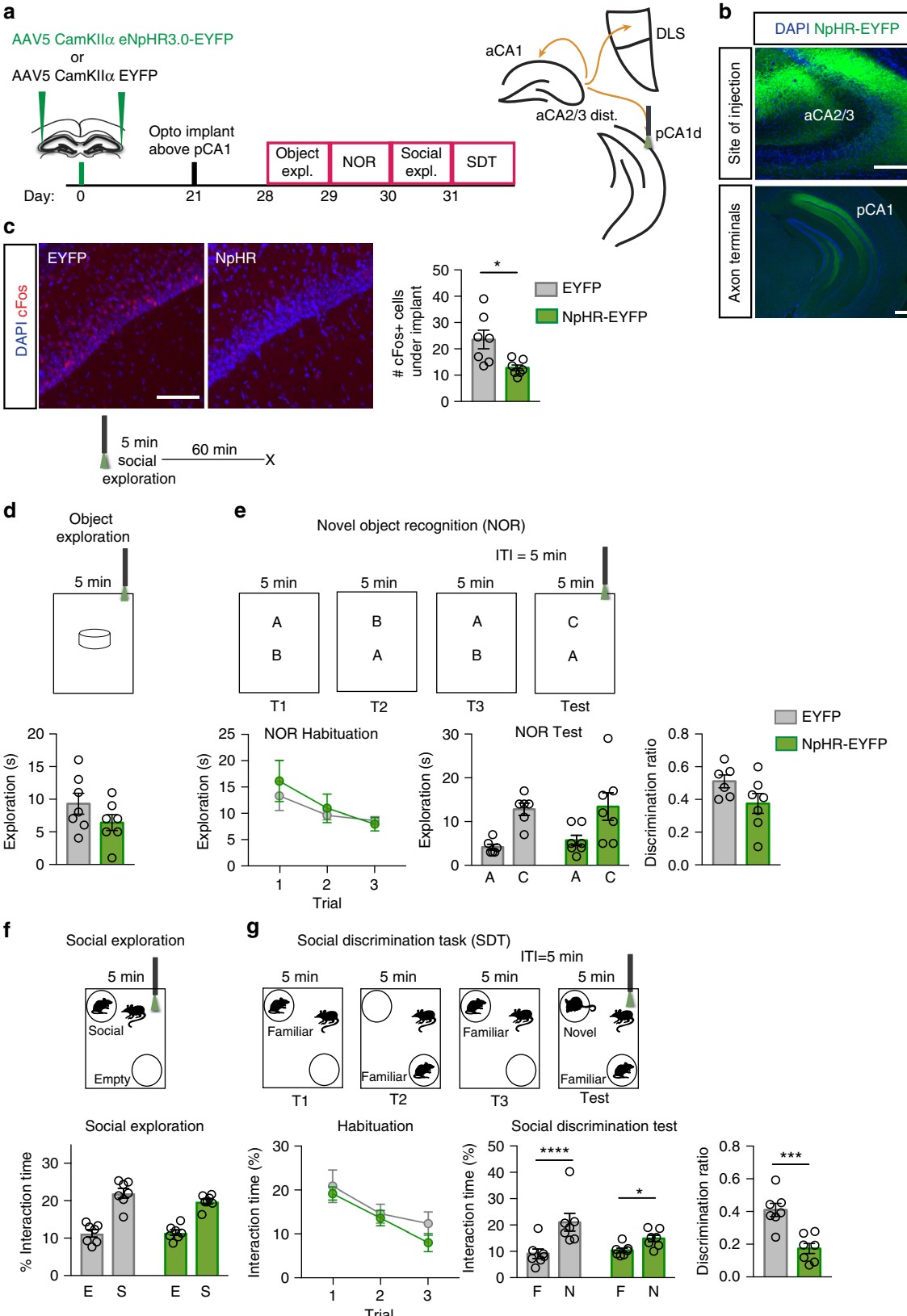

discrimination of social stimuli. The modest levels of discrimination of social stimuli that were persistent following optogenetic attenuation of aCA2/CA3 projections to pCA1 may reflect insufficient illumination of the large termination zone in pCA1d, incomplete silencing, and/or lack of targeting just the Oxtr-expressing neurons in aCA2/3. Nevertheless, these results are consistent with a recent study implicating the pCA1-NAcc pathway in mediating discrimination of social stimuli. Whether this pathway or other pCA1 projections to prefrontal cortex, hypothalamus, or amygdala relay Oxtr-dependent computations underlying social recognition in anterior hippocampus to guide social behavior remains to be determined. Our findings that aCA2/CA3 projections to aCA1 mediate discrimination of non-social, but not social, stimuli are consistent with the role of aCA1 in object recognition[27,28]. Thus, social and non-social information maybe differentially routed out of aCA2/CA3 to aCA1 or pCA1. Differential routing may be dependent on engagement of OT signaling in microcircuits in aCA2/CA3 or differences in projection patterns of Oxtr-expressing aCA2/CA3 neurons to aCA1 and pCA1.

In summary, our studies begin to illuminate the circuit mechanisms and neural pathways by which hippocampal Oxtrs contribute to social recognition. Oxtrs appear to recruit basic circuit mechanisms supporting pattern separation such as population based coding to minimize interference between similar social memories. These cognitive computations are relayed from the anterior to posterior hippocampus via an aCA2/CA3-pCA1 pathway to guide social behavior. Thus, disruptions in DG-CA2/CA3 circuitry may contribute to social memory impairments seen in autism and other psychiatric disorders[1–3,68–70].

## Methods

**Mouse lines and animal care**. Two strains of mice were used in these experiments. Multiplex FISH and optogenetic terminal-specific silencing experiments were carried out in C57Bl/6J mice. Oxtr conditional knockout mice ($Oxtr^{f/f}$) rederived at Jackson Labs were utilized for viral-mediated recombination of Oxtrs[35](Jackson Laboratories stock 008471). All experiments were carried out in adult male mice (8 weeks old). All mice were housed three to five per cage in standard laboratory conditions under a 12 h light–dark cycle (0700–1700 h) with ad libitum access to food and water. Behavioral testing occurred during the light cycle. All experiments were carried out in accordance with procedures approved by the Institutional Animal Care and Use Committee at the Massachusetts General Hospital in accordance with NIH guidelines.

**Multiplex fluorescence in situ hybridization**. FISH was carried out using ACDBio RNAscope multiplex fluorescence assay. Probes were obtained as following: C1: PV (Mm-Pvalb: 421931), Cre (CRE: 312281), C2: Oxtr (Mm-Oxtr-C2: 402651-C2), C3: Gad1 (Mm-Gad1-C3: 400951-C3), and SST (Mm-Sst-C3: 404631-C3). Brain tissue was obtained from age-matched 8-10 week-old male and female C57BL/6J mice and immediately fresh-frozen in optimal cutting temperature (OCT) medium on dry ice. For the assessment of Oxtr knockout, $Oxtr^{+/+}$ and $Oxtr^{f/f}$ mice were injected in aDG or aCA2/CA3 with AAV-CRE (AAV-hSyn Cre-GFP, diluted 1 : 100, 0.3 μl bilaterally, in aDG and AAV₉-CamKII-Cre-GFP 1 : 5 dilution, 0.3 μl bilaterally in aCA2.3). Twenty-micrometer sections were obtained using a cryostat (Leica, Germany) and stored at − 80 °C. Before RNAscope protocol, sections were incubated at − 20 °C for 1 h. RNAscope protocol was followed as indicated in user manual with adjustments as mentioned here. Briefly, sections were fixed with 4% chilled paraformaldehyde (PFA) for 30 or 60 min and rinsed in

1× phosphate-buffered saline (PBS) for 5 min. Sections were then dehydrated in four 5 min dehydration steps in 50%, 75%, 100%, and 100% ethanol, respectively. Tissue was digested with Protease III for 30 min at room temperature and rinsed in 1 × PBS. Oxtr and Gad1 probes were diluted 1 : 50 in probe diluent and pipetted onto each slide. Oxtr and SST probes were diluted 1 : 50 in PV C1 probe and pipetted onto each slide. For assessment of OxtR knockout, Oxtr probe was diluted 1 : 50 in Cre C1 probe. Probe hybridization took place for 2 h at 40 °C and slides were then rinsed in 1 × wash buffer. Sections were then incubated at 40 °C in Amp1 for 30 min, Amp 2 for 15 min, Amp3 for 30 min, and Amp4 for 15 min, with 1 × Wash Buffer rinses between each incubation. Sections were finally mounted with DAPI (4',6-diamidino-2-phenylindole) Fluoromount and stored at − 20 °C.

**Adeno-associated viruses (AAV)**. AAV₅-CaMKIIα-eYFP ($6 \times 10^{12}$ particles per mL) and AAV₅-CaMKIIα-eNpHR3.0-eYFP ($3 \times 10^{12}$ particles per mL) viruses were generated by and obtained from the University of North Carolina Vector Core (Chapel Hill, NC). AAV₉-CamKIIα-Cre-GFP ($1.07 \times 10^{13}$ particles per mL), AAV₉-CamKIIα-GFP ($6.27 \times 10^{13}$ particles per mL) and AAV₉-hSyn-Cre-GFP ($11.7 \times 10^{13}$ particles per mL) were generated by and obtained from the University of Pennsylvania Vector Core (Philadelphia, Pennsylvania).

**Stereotactic surgery**. Viral injections: adult mice (8 weeks old) were maintained under standard housing conditions. Mice were given carprofen (5 mg kg⁻¹ subcutaneously) before surgery and 24 h later to minimize discomfort. Mice were anesthetized with ketamine and xylazine (10 and 1.6 mg mL⁻¹, intraperitoneally (i.p.)) and placed in a stereotaxic frame (Stoelting). Small holes were drilled at each injection location and injected with a Hamilton microsyringe at a rate of 0.1 μl min⁻¹. For aDG KO experiment, 0.2 μl 1 : 1,000 AAV9-hSyn-Cre-GFP or 1 : 50 AAV9-CamKIIα-GFP was injected bilaterally (AP − 1.9 mm, ML ± 1.1 mm, DV − 2.6 mm from Bregma) into Oxtr f/f mice. For aCA2/CA3_distal KO experiments, 0.3 μl 1:50 AAV9-CamKIIα-Cre-GFP or 1 : 50 AAV9-CamKIIα-GFP was injected bilaterally (AP − 1.9 mm, ML ± 2.55 mm, DV − 2.3 mm from Bregma) into Oxtr f/f mice. For optogenetic silencing experiments, 0.3 μl undiluted AAV5-CaMKIIα-eNpHR3.0-eYFP or AAV5-CaMKIIα-eYFP was injected bilaterally into aCA2/CA3_distal. After completion of the injection, the needle was left on the site of injection for 5 min, raised 0.2 mm, and left on the site for an additional 5 min, to allow diffusion of virus at the injection site and then slowly withdrawn. The skin incision was closed carefully using nylon sutures. Animals were monitored during recovery.

Optogenetic implants: mice were surgically implanted with fiber optic cannulas 3–4 weeks following injection of AAV5-CaMKIIα-eYFP or AAV5-CaMKIIα-eNpHR3.0-eYFP and behavioral experiments started 1 week after surgery. Mice were implanted bilaterally with chronically dwelling optical fibers targeted to the anterior CA1 (AP − 1.9 mm, ML ± 1.5 mm, DV − 1.2 mm from Bregma), pCA1 (AP − 3.35 mm, ML ± 3.1 mm, DV − 1.7 mm from Bregma), or DLS (AP + 1.15 mm, ML ± 1.0 mm angle ± 10°, DV − 2.5 mm from Bregma). Optical fibers were secured with one cranial screw and dental cement. After surgery, mice were returned to their home cage and monitored until recovery from surgery.

**Construction of optical fibers**. Two hundred micrometer core, 0.37 numerical aperture multimode fiber (ThorLabs) was threaded through and glued with epoxy to a 230 μm core stainless steel and zirconia multimode ferrule (Fiber Instrument Sales and Precision Fiber Products), polished, and cut for implantation. Optical patch cables were generated the same way, with the free end connected to a multimode FC ferrule assembly for connecting to a 1 × 2 Optical rotary joint (Doric lenses). The other end of the rotary joint was connected via a patch cable to a 561 nm laser diode (OEM laser systems) via a non-contact style laser to fiber coupler (OZ optics).

**In vivo laser delivery**. Age-matched, genotyped-matched male mice (3–4 months old) were used for all behavioral experiments. Mice were kept in a quiet room for at least 1 h before testing. Behavioral tests took place under bright lighting conditions (700 lux) and performed in the following order: single object exploration (day 1), novel object recognition (day 2), social exploration (day 3), and social discrimination (day 4). Before each experiment, mice were bilaterally attached to the patch cables via a zirconia sleeve and allowed to recover for 2–3 min in a transition

**Fig. 6** Optogenetic attenuation of aCA2/CA3_distal outputs to pCA1 impairs discrimination of social, but not non-social, stimuli. **a** Schematic illustrating viral injection, optogenetic implants and behavioral testing timeline. **b** Representative images of site of injection of eNpHR3.0 virus in aCA2/CA3_distal cell bodies (top) and corresponding axon terminals in pCA1 (bottom). Scale bars, 200 μm (top) and 500 μm (bottom). **c** Representative images and quantifications of cFos immunoreactivity in pyramidal layer of pCA1 during optogenetic silencing (EYFP: n = 7, NpHR: n = 7). Scale bar, 100 μm. **d** Behavioral schematic (top) and quantification (bottom) of single object exploration (EYFP: n = 7, NpHR: n = 7). **e** Behavioral schematic (top) and quantification (bottom) of novel objection recognition (EYFP: n = 6, NpHR: n = 7). Quantifications are displayed as Habituation (trials 1–3), Test (trial 4), and discrimination ratio (trial 4). Laser was on during trial 4 only. **f** Behavioral schematic (top) and quantification (bottom) of social exploration test (EYFP: n = 6, NpHR: n = 7). **g** Behavioral schematic (top) and quantification (bottom) of social discrimination task (EYFP: n = 6, NpHR: n = 7). Quantifications are displayed as Habituation (trials 1–3), Test (trial 4), and discrimination ratio (trial 4). Laser was on during trial 4 only. All data are displayed as mean ± SEM

cage similar to their home cage. The patch cables were interfaced to an FC/PC rotary joint (Doric lenses), which was attached on the other end to a laser diode (561 nm). The light power at the end of the fiber tip was 10–15 mW. At the end of each behavioral experiment (5–7 weeks following viral surgery), mice were analyzed for viral and fiber optics placement and mis-targeted animals were excluded from the analysis. In aCA2-CA3-CA1 attenuation experiment, animal numbers varied across behavioral tasks because of loss of optogenetic implants during course of testing and two behavioral cohorts were utilized, with the second cohort being used for a context-updating task instead of social discrimination after social exploration. Randomly selected videos were scored by two different investigators.

**Single object exploration**. Mice were placed an arena filled with bedding (45 cm long, 15 cm high and 30 cm wide) with a single novel object for 5 min. Sessions were video-recorded and videos were manually scored for object exploration (when an animal's snout was 2 cm or less from the object) by an investigator blind to animal treatment identity. For optogenetic experiments, light of 561 nm was turned on for the full duration of the session.

**Novel object recognition**. Mice were placed in an arena filled with bedding (45 cm long, 15 cm high and 30 cm wide) with two distinct objects (A and B) for three 5 min sessions, with a 5 min intertrial interval. Mice habituated to the objects during the three training sessions and one of the objects was subsequently replaced with a novel object (C) in session four. Objects were counterbalanced across subjects and object positions were counterbalanced across trials. The objects that were selected for testing comparable levels of exploration based on exploration levels in pilot experiments. Sessions were video-recorded and videos were manually scored for object exploration (when an animal's snout was 2 cm or less from the object) by an investigator blind to object identity and animal treatment identity. For optogenetic experiments, light of 561 nm was turned on only during trial 4.

**Social exploration**. Mice were placed in an open plexiglass arena (41 cm × 41 cm, Kinder Scientific) for 5 min with two pencil-wire cups placed on opposing corners of the arena. Habituation to the chamber took place one day before the start of behavioral testing. One cup remained empty, whereas the other contained a novel adult male C57Bl/6J mouse. Sessions were video-recorded and videos were manually scored for exploration of the social cup and empty cup (direct snout-to-cup contact, with at least two paws on the ground—time spent climbing on the cup was not included) by an investigator blind to animal treatment identity. Stimulus mice were trained to sit in pencil-wire cups for three 15 min sessions several days before behavioral testing. For optogenetic experiments, light of 561 nm was turned on for the full duration of the session.

**Social recognition (habituation and discrimination)**. The same plexiglass arena and pencil-wire cup set-up as described for social exploration was used. Mice were placed in the arena for three 5 min sessions with an empty cup and a novel adult male C57Bl/6J mouse, with a 5 min intertrial interval. Mice habituated to the stimulus during the first three sessions, rendering it "familiar." During trial 4, a novel adult male mouse was placed in the opposing cup and subjects were tested for discrimination between the novel and familiar mouse. Social stimuli were counterbalanced across subjects and the position of the familiar social stimulus was counterbalanced across trials. Sessions were video-recorded and videos were manually scored for interaction with the novel vs familiar mouse (direct snout-to-cup contact, with at least two paws on the ground—time spent climbing on the cup was not included) by an investigator blind to animal treatment identity. Stimulus mice were trained to sit in pencil-wire cups for three 15 min sessions several days before behavioral testing. For optogenetic experiments, light of 561 nm was turned on only during trial 4. Videos were scored by two different investigators.

**aDG genetic deletion cohorts**. For Fig. 2 and the corresponding Supplementary Fig. 3, number of animals is variable across behaviors, because two different cohorts were used. The first cohort ($n = 7, 4$) was used for all behaviors and cFos analysis. The second cohort ($n = 4, 2$) was used to increase the $N$ for the four behaviors in the main Figure, but was not used for anxiety (OFT and EPM) or cFos analysis.

**catFISH assay**. The catFISH assay was performed as previously described[47]. Briefly, adult male *Oxtr f/f* mice received bilateral stereotactic injections of AAV9-CamKIIα-Cre-GFP diluted at 1 : 50 into aCA2/CA3$_{distal}$ 10 days before behavior, as described above. Animals experienced two 5 min behavioral episodes 25 min apart and brains were isolated and flash frozen in isopentane on dry ice immediately after the second episode. An intronic *c-fos* probe containing the entire first intron of the Fos gene[54] (generous gift of Dr. Dayu Lin) and a full-length *c-fos* cRNA probe were used to detect nuclear and cytoplasmic c-fos transcripts, respectively. Nuclear c-fos was defined as puncta colocalizing with DAPI labeling, whereas cytoplasmic labeling was defined as c-fos-positive labeling surrounding the nucleus. CA3$_{proximal}$ comprised an region of interest (ROI) with a volume of 788,546 cubic micrometers

per section and contained on average 521.67 DAPI$^+$ cells ($n = 3$). CA3distal comprised an ROI with a volume of 1,216,503 cubic micrometers per section and contained an average of 554.67 DAPI$^+$ cells ($n = 3$). CA2 was not quantified due to too few cFos$^+$ foci for analysis (on average two cFos$^+$ cells per section). Representative images of each condition are included in.

**Histology and immunohistochemistry**. Mice were anesthetized with ketamine (100 mg kg$^{-1}$ i.p.) and xylazine (3 mg kg$^{-1}$ i.p.), and transcardially perfused with 10 mM PBS, pH 7.5 at 4 °C, followed by a fixative solution containing 4% PFA in PBS. Brains were post-fixed in 4% PFA overnight at 4 °C and cryo-protected for 72 h in 30% sucrose at 4 °C before freezing in OCT on dry ice. Coronal sections were obtained at 35 µm using a cryostat (Leica, Germany) in six matched sets. Sections were stored in 1 × PBS with 0.01% Azide at 4 °C. Floating sections were used to perform immunohistochemical labeling. Briefly, sections were washed in 1 × PBS, incubated for 15 min at room temperature in 1 × PBS containing 0.3% Triton X-100, blocked in 1 × PBS containing 0.3% Triton X-100 and 10% Normal donkey serum for 1 h at room temperature, and incubated in primary antibodies in blocking buffer with shaking overnight at 4 °C. Primary antibodies were used as follows: GFP (rabbit anti-GFP, Life Technologies A11122, 1 : 500, Antibodyregistry.org: AB_221569); goat anti-GFP, Novus Biologicals, NB100-1770, 1 : 500, (Antibodyregistry.org: AB_10128178), cFos (Rabbit, Santa Cruz, 1 : 2,000, Santa Cruz SC52, Antibodyregistry.org: AB_2106783), RGS14 (Mouse, UC Davis/NIH NeuroMab, 1 : 400). The following day, sections were washed three times in 1 × PBS and incubated with fluorescent-label-coupled secondary antibodies (Jackson Immunoresearch, Donkey anti-Rabbit, Goat, or Mouse, 488-, Cy3-, or Cy5-coupled, 1 : 500) for 2 h at room temperature.

**cFos and perfusion**. Mice were exposed to a novel social stimulus in a novel arena filled with bedding for 5 min and promptly returned to home cage. Mice were then anesthetized with ketamine (100 mg kg$^{-1}$ i.p.) and xylazine (3 mg kg$^{-1}$ i.p.), and transcardially perfused 60 min post exposure, as described above in "Histology and immunohistochemistry." For optogenetic silencing cFos experiments, mice were given 3–5 min to acclimate after being bilaterally attached to a laser patch cord, before the laser was turned on and social exposure began.

**Image acquisition and analysis**. For RNAscope multiplex FISH: × 20, × 40, or × 60 confocal z-stack images were obtained with a Nikon A1R Si confocal laser. Images were acquired at 1,024 resolution as 10 µm z-stacks with a step size of 2 µm. For colocalization analysis, cells were manually identified as *Oxtr$^+$*, *Gad1$^+$*, *SST$^+$*, and/or *PV$^+$* or double positive, and expressed as a percentage of *Oxtr$^+$* cells (for DG) or as a percentage of *SST$^+$*, or *PV$^+$* cells (for CA2/CA3). For *Oxtr* knockdown analysis, all DAPI nuclei per image were counted (for DG injections: cells in the hilus, excluding granule cell layer and CA3, were counted, whereas for CA2/CA3 injections cells in the CA2/CA3 principal cell layer were counted), and cells were manually identified as *Oxtr$^+$*, *Cre$^+$*, and/or *OxtR$^+$CRE$^+$* double positive. The *Oxtr$^+$* population was expressed as a percentage of total DAPI cells. For *Oxtr* mean intensity calculated for CA regions, mean intensity was measured for the region comprised by the principal layer. A similar sized region from background area was measured for intensity and subtracted from mean *Oxtr* intensity, in order to control for differences in background.

For cFos analysis: × 10 bilateral images were acquired using a Nikon epifluorescence microscope. cFos+ cells were manually quantified as those with intensity higher than background in the aDG granule cell layer, pCA1 pyramidal cell layer directly under the implant, and DLS. For aDG, both blades were assessed. Data were expressed as # cFos+ cells per section. All quantifications and analyses were completed by an experimenter blind to treatment condition.

For catFISH: confocal z-stack images using a × 60 oil objective (1.0 µm step size) were acquired along the aCA2/CA3 pyramidal layer using a Nikon A1R Si confocal laser, a TiE inverted research microscope, and NIS Elements software. Image analysis was performed with NIS Elements software. Four to eight hemisections were scanned per mouse. Images were manually counted for c-fos transcript labeling. Data are presented as cells/section. Quantifications were performed by an investigator blind to treatment condition and two different investigators. Details of OFT, EPM, and pharmacological blockade of Oxtrs can be found in Supplementary Note 1.

**Statistics**. Statistical analysis was carried out using GraphPad Prism v7 software. Data (mean ± SEM) were analyzed using paired or unpaired two-tailed Student's *t*-test, to compare two groups. Behavioral data were analyzed using mixed factor two-way ANOVA with repeated measure followed by Bonferroni's multiple comparisons test when the interaction, *p*-value was < 0.05. Detailed statistical analyses can be found in Supplementary Table 1. In all cases, significance was set at $p < 0.05$.

**Data availability**. All relevant data are available from the authors.

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

## Acknowledgements

We thank members of the Sahay lab for their comments on the manuscript. A.S. acknowledges support from US National Institutes of Health Biobehavioral Research Awards for Innovative New Scientists (BRAINS) 1-R01MH104175, NIH-NIA 1R01AG048908-01, NIH-NIMH 1R01MH111729-01, Ellison Medical Foundation New Scholar in Aging, Whitehall Foundation, Inscopix Decode award, NARSAD Independent Investigator Award, Ellison Family Philanthropic support, Blue Guitar Fund, Harvard NeurodiscoveryCenter/MADRC Center Pilot Grant Award, HSCI Development grant, and HSCI seed grant.

## Author contributions

T.R., K.M.M., and A.B. carried out experiments and analyzed data. A.V. contributed reagents and shared resources. T.R. and A.S. co-developed the concept and wrote the manuscript. A.S. conceived and supervised all aspects of the project.

## Additional information

**Competing interests:** The authors declare no competing financial interests.

