## [Peer Review File · Nature Communications]

Reviewers' expertise:

Reviewer #1: neural circuits underlying social behaviors, systems neuroscience approaches;

Reviewer #2: oxytocin signaling;

Reviewer #3: hippocampus, plasticity.

Reviewers' comments:

Reviewer #1 (Remarks to the Author):

Hippocampal oxytocin receptors are necessary for discrimination of social stimuli

Tara Raam, Kathleen M. McAvoy, Antoine Besnard, Alexa Veenema, and Amar Sahay

Comments:

This study by Raam and colleagues identifies oxytocin receptor expressing populations in the hippocampus and uses a number of anatomical, molecular, and optogenetic manipulations to characterize their role in encoding social recognition. Previous studies have identified high levels of oxytocin receptor expression in subregions of the hippocampus using transcript tagging, antibody labeling of protein, as well as retrograde tracing and anatomical mapping techniques 1–4. Previous studies have also identified plasticity of excitatory synapses evoked by bath application of oxytocin in the CA2 and CA3, but not CA1 5. At the same time, acting on fast-spiking interneurons in the CA1, oxytocin has been shown to enhance inhibitory spike transmission 6. In vivo and behavioral studies have also implicated the hippocampus, especially the CA2 and ventral CA1, in the regulation of social memories 7–9, although they did not directly address the role of oxytocin. The current study by Raam, et al., represent the convergence of these previous insights, and offer novel insights into the role of oxytocin, acting in the hippocampus, in regulating social memories. Nevertheless, the conclusions would be significantly strengthened by the additional experiments outlined below.

Major:

1. Experiment 1: Previous studies have suggested an important role for presynaptically localized OTR's 1,3,6. Consistent with these findings, in figure 1 and S1, Raam et al show that in both the ventral and dorsal CA2/3, OTR mRNA is densely expressed in the pyramidal cell layer, but does not take on a distribution consistent with cell body localization (in contrast to the CA1). Since previous studies have demonstrated that presynaptic terminals are capable of protein synthesis 10, it is possible that OTR mRNAs localized to CA2/3 are not made by hippocampal pyramidal neurons, but are instead made by inputs to the CA2 (e.g. the hypothalamus 11 which contains oxytocin neurons themselves producing autoreceptors 12). Although it is perhaps beyond the scope of this paper to ask for EM level characterization to differentiate these possibilities, electrophysiological characterization of miniature excitatory postsynaptic frequency and amplitude, combined with anterograde tracing from the hypothalamus (or other input regions) could clarify these possibilities, which dramatically alter the interpretation of the results, especially the cFOS studies in

Figure 4, which are an indirect measure of neuronal activity in response to social stimuli.

2. Experiment 2: Although the authors would like to claim that molecular ablation and optogenetics experiments effectively localize their effects to hippocampal CA2/3 pyramidal cells, for reasons that are unclear, they have carried out their experiments using the AAV9 serotype, which has previously been shown to be taken up by presynaptic terminals and retrogradely transported 13,14. Thus the effects reported in figures 3, 4, and 6, may in fact reflect ablation or silencing of presynaptic OTRs originating outside the hippocampus, particularly in light of the distribution pattern of OTR mRNA in Figures 1 and S1 (discussed above). Thus the conclusions the authors draw would be significantly strengthened if the authors could show that these effects were absent following Cre or halorhodopsin expression using anterograde viruses into the hypothalamus and other input regions to the CA2/3, or retrograde viruses (e.g. CAV2, RbV, or pseudotyped LV) into the CA2/3.

Minor:

1. It is not clear in Figure 2 why the authors used two different promoters (e.g. hSyn or CamKIIa) to drive expression of Cre-GFP versus control GFP?
2. The authors argue that habituation is impaired only following OTR knockdown in dCA2/3 (figure 3f) but not dDG (figure 2g), but in the second trial habituation looks significantly impaired for dDG knockdown, and the effect is much less impressive in dCA2/3.
3. In figure 4, the authors make a point of differentiating between CA2, CA3b and CA3c, but do not describe the functional or anatomical basis for defining these subregions.

1. Dölen, G., Darvishzadeh, A., Huang, K. W. & Malenka, R. C. Social reward requires coordinated activity of nucleus accumbens oxytocin and serotonin. *Nature* 501, 179–84 (2013).
2. Yoshida, M. et al. Evidence that oxytocin exerts anxiolytic effects via oxytocin receptor expressed in serotonergic neurons in mice. *J. Neurosci.* 29, 2259–71 (2009).
3. Mitre, M. et al. A Distributed Network for Social Cognition Enriched for Oxytocin Receptors. *J. Neurosci.* 36, 2517–2535 (2016).
4. Knobloch, H. S. S. et al. Evoked axonal oxytocin release in the central amygdala attenuates fear response. *Neuron* 73, 553–66 (2012).
5. Pagani, J. H. et al. Role of the vasopressin 1b receptor in rodent aggressive behavior and synaptic plasticity in hippocampal area CA2. *Mol. Psychiatry* (2014).
doi:10.1038/mp.2014.47
6. Owen, S. F. et al. Oxytocin enhances hippocampal spike transmission by modulating fast-spiking interneurons. *Nature* 500, 458–62 (2013).
7. Alexander, G. M. et al. Social and novel contexts modify hippocampal CA2 representations of space. *Nat. Commun.* 7, 10300 (2016).
8. Okuyama, T., Kitamura, T., Roy, D. S., Itohara, S. & Tonegawa, S. Ventral CA1 neurons store social memory. *Science* (80-.). 353, 1536–41 (2016).
9. Hitti, F. L. & Siegelbaum, S. A. The hippocampal CA2 region is essential for social memory. *Nature* 508, 88–92 (2014).
10. Akins, M. R. et al. Axonal ribosomes and mRNAs associate with fragile X granules in adult rodent and human brains. *Hum. Mol. Genet.* 26, 192–209 (2017).
11. Cui, Z., Gerfen, C. R. & Young, W. S. Hypothalamic and other connections with dorsal CA2 area of the mouse hippocampus. *J. Comp. Neurol.* 521, 1844–66 (2013).
12. Freund-Mercier, M. J., Stoeckel, M. E. & Klein, M. J. Oxytocin receptors on oxytocin

neurones: histoautoradiographic detection in the lactating rat. *J. Physiol.* 480 (Pt 1, 155–61 (1994).

13. Löw, K., Aebischer, P. & Schneider, B. L. Direct and retrograde transduction of nigral neurons with AAV6, 8, and 9 and intraneuronal persistence of viral particles. *Hum. Gene Ther.* 24, 613–29 (2013).

14. Cook-Snyder, D. R., Jones, A. & Reijmers, L. G. A retrograde adeno-associated virus for collecting ribosome-bound mRNA from anatomically defined projection neurons. *Front. Mol. Neurosci.* 8, 56 (2015).

Reviewer #2 (Remarks to the Author):

The manuscript of Tara Raam and colleagues is focused on an important and novel topic: the functional role of oxytocin (OT) signaling in the dorsal hippocampus and its contribution in discrimination of social stimuli. Employing knock-in OTR-Cre mice, the authors showed the expression of OTR in GABA-ergic neurons (86%) and in a smaller population of principal neurons. The deletion of OTR in excitatory OTR+ cells resulted in a deficit in social discrimination tasks, but did not affect object exploration and novel object exploration. In line with this finding, the authors showed that principal cells lacking OTR were equally reactivated (judged by subcellular c-fos distribution, catFISH) after repeated exposure to known and novel mice. Finally, using optogenetics, the authors showed that the output of pyramidal OTR+ neurons of dCA2/CA3 to dCA1 is critical for social discrimination.

Although the experiments are straightforward and results are well documented, I would like to raise several concerns:

1) In Figure 1, the authors showed an overlap of OTR expression with GAD1 both within dorsal and ventral CA2/3. The presence of even a minor population of inhibitory cells receptive to OT in this region might have indeed an important role not only in intra-hippocampal circuitry, but also in hippocampal output structures. Taking into account the growing evidence showing unique properties of CA2 interneurons, it would be reasonable to show the type of interneurons expressing OTR in the CA2/CA3 region. A feasible solution of this question, in my opinion, is to employ staining of calcium-binding proteins to identify subclasses of local interneurons expressing OTR.

2) The authors claim that the OTR expression in dCA1 is absent. Since the effects of TGOT on hippocampal oscillatory activity in CA1 were described in the literature, and previous *in situ* studies indeed showed the presence of OTR mRNA in the principal cell layer of CA1, it would be important to demonstrate OTR mRNA expression in mouse brain planes containing both dorsal and ventral CA2/3 (Br level: -5; -5,7).

3) In the experiments showing impairment of social stimuli discrimination after acute blockade of OTR in CA2/3, the authors used OTR antagonist L-368, 899 Hydrochloride. How did the authors determine the spread of OTR-A? It is difficult to imagine that infusion of OTR-A will affect only the dCA2/CA3 as the compound is characterized by great potency and extremely rapid brain penetration.

4) The authors should confirm that the rAAV9, equipped with CamKII, is exclusively expressed in principal cells. From my experience, infection with CamKII-rAAV1/2 leads to expression of GFP also in interneurons.

5) To confirm a critical role of the OTR dCA2/3 – dCA1 circuit in the modulation of social discrimination, the gain of function experiment (Chr2) should be added as an essential control.

Reviewer #3 (Remarks to the Author):

Raam et al. review

In the manuscript by Raam, et al., the authors detail the results of experiments aimed at clarifying the role of hippocampal oxytocin receptors in regulating the circuitry underlying social behavior. A more complete understanding would also include 1) studies on the cellular effects of oxytocin in the neurons studied (i.e. is it excitatory?), 2) consideration of the PVN axons into the hippocampus, and/or 3) more detail on how the DG interneurons impact the circuit. Nevertheless, within the scope of the study as it is presented, the manuscript is a valuable contribution to the literature in that it addresses both transverse (dorsal) and longitudinal (dorsal-ventral) aspects of the hippocampal circuitry, as well as three major outputs of the oxytocin-expressing principle neurons in hippocampus. The experiments were apparently well designed and executed, and the manuscript and figures were generally nice to read. My only major concerns are those related to the interpretation of the data, both in terms of overstating them, or more concerning in some cases, overt misinterpretation of the anatomical areas involved. Most of the concerns though, listed below, can be addressed without new experiments.

1. Throughout, the authors refer to the area of densest Oxtr expression as area CA2/3, yet the authors appear to also be aware that there are huge differences between high Oxtr expression levels in CA3a (near CA2), and CA3b and c, which are quite a bit lower. Therefore it would be appropriate that the authors distinguish CA3a and use a more accurate description throughout: CA2/CA3a (vs. CA2/CA3). Second, it would also seem worthwhile to address why CA3b, with its intermediate levels of Oxtr (at best), is grouped with CA3a, and not CA3c, for the catFISH experiments in Fig 4, even if was simply for technical reasons relating to the injections.

2. In a related topic, the authors are using RNAscope, so they should be able to accurately quantify the density of transcripts in each region for figure 1: CA1, CA2/CA3a, and CA3c (ideally also CA3b separately, but this would require some markers or clear anatomical features distinguishing the regions). Also, for figure 1b and c, similar pie charts as in a).

3. The authors refer to a vCA1d, which is really just dorsal CA1. Just because it is further back in a coronal section does not make it in any way 'ventral'. At best, it is mid-way along the hippocampal axis, but at that (horizontal) level, the dentate is clearly a 'V' shape (vs. a 'U' seen in the ventral hippocampus). If the authors have horizontal sections showing dCA2/CA3a projections in vCA1, they would also be very useful to see (perhaps in Fig 5b). Again- some quantification of the Oxtr expression in ventral CA1 would be required if the

authors are to claim Oxtr are higher in the ventral CA1. Because determining subfields in ventral hippocampus is very difficult in coronal sections, the authors would ideally quantify Oxtr levels in horizontal sections. At least, though, they should use an adjacent section stained for a CA1 marker such as wfs1 to confirm what they are looking at is actually CA1. It looks like they could possibly be looking at the subiculum, depending on the sections.

4. The authors focus their viral recombination/Oxtr deletion injections in either dentate gyrus or dorsal CA2/3a, but because they are not genetically targeted, the authors need to provide more images, including a panel of lower magnification images for the readers to get a better understanding of where exactly those injections were landing, and whether the cre is specifically targeted. Drawings, such as in Fig S1b and c, would also be helpful in many cases. Specifically, with regard to the dCA2/3a injection, is Figure 3b representative (it appears to largely be in CA2)? Figure 5b, on the other hand, would be improved by showing multiple planes of sections at low power, and indicating the injection site and projections (in higher power). In addition, there is some concern that the DG injections were not specific, given a peculiar affinity CA2 neurons have for AAVs. Without genetically targeted Cre expression, the documentation of the injections should be more rigorous and inclusive. The additional images would be appropriate for supplementary material if not the main figures.

5. For the optogenetic silencing experiment in the dCA1 projection target from dCA2/3a (Figure 5) the authors don't show cFos data to confirm silencing (which they did do in the DLS and vCA1 experiments). Why not? The silencing of CA2/CA3a may very well increase fos, given one idea in the field that CA2 output has a net inhibitory effect on at least some neurons in CA1, but the reasons for why this data was excluded should at least be given.

6. Ideally, if the authors still have the sections from the deletion experiments, they should confirm actual knockdown, either with RNAscope or with an antibody (if they can acquire one, from Mitre, et al., for example).

Minor:

1. Please be consistent in using the correct nomenclature for rodent genes (italics, first letter of symbol in caps) vs proteins (not italics, all caps).
2. Pg 3, first paragraph second sentence: this is not the definition of global remapping. Two non-overlapping, sparsely encoded ensembles is the proposed mechanism behind pattern separation. Global remapping refers to modifying an existing ensemble [by rate or field changes] to reflect updated context. See Colgin et al 2008 TINs review currently cited as #48.
3. same paragraph- It is important to distinguish when the species differs in the studies used for the rationale.
4. Last sentence pg 5: Object discrimination was not tested with pharmacology
5. Pg 6, "well-characterized Schaffer collateral pathway from dCA2/CA3-dCA1". As this projection populates primarily the stratum Oriens in CA1, I don't think the dCA2->CA1 projection is widely considered to be a part of the Schaffer collaterals, and anything from CA2 is far from being 'canonical'. Therefore I suggest the authors use a less controversial description such as, "we first targeted the projection from dCA2/CA3a to dCA1."
6. In general, the authors should be wary of how they use of the term "retrieval" in describing the results of the novel object recognition experiments. Given that the time point for testing is 30 min post first exposure/encoding, the authors are not testing retrieval in the way it is more typically performed (i.e. more usual is 24 hours post-encoding to test

long term memory retrieval). Because of this different methodology, the reader is unable to extrapolate the results to the large amount of literature on this subject. Furthermore, the possible plasticity mechanisms/brain regions involved at this time point are not clear or not discussed. The authors should consider discussing this distinction in the context of OXTR signaling as well as their reasoning for choosing this time point.

7. It would be important to discuss why CA2 was not assessed in the catFISH experiments. This is likely due to the fact that cFos foci are very small in CA2 and so are difficult to quantify (Alexander, et al.), but this should be explicitly stated given that Oxtrs were knocked out of CA2/CA3a and no affect was found for the CA3ab region. Furthermore, it is unclear why there would be an effect in CA3c. This should also be discussed.

8. There is no mention in the discussion of the fact that attenuation of dCA2/3 axons in vCA1 (using the dCA2/3 eNpHR3) showed no significant difference in the social discrimination test itself (Fig 6g). Is the discrimination ratio difference enough to state the importance? If the eNpHR positive fibers only represent a fraction of the dCA2/3 to vCA1 fibers that produced the robust effects in the Oxtr deletion experiment (Fig. 2g), that should be addressed.

9. How does the retrieval phase of the social discrimination task (p5, during pharm block of OXTRs) compare with the test phase of the task (p7, during circuit attenuation)? I'm assuming these are the same periods of the same task.

10. The authors should discuss their findings in the context of a recent oxytocin receptor paper Ripamonti, et al. 2017 and/or the sexual dimorphism seen in Oxtr distribution (Mitre, 2016)

Methods:

11. Were multiplex bacterial negative controls performed alongside the smFISH experiments to appropriately control for background during image acquisition? If so, describe.

12. For catFISH experiments, given the differences in cell number due to difference in section/angle and CA3ab vs CA3c, it would be preferable to present the data as % of cells instead of (or in addition to) # of cells per section. At the very least, provide the dimensions of the ROI assessed (i.e. 500microns squared) and provide an average number of cells assessed per region so that percentages can be inferred by a knowledgeable reader.

13. For DG FOS histology experiments, please elaborate whether both blades were counted/assessed.

14. Fig 2. The time-line for FOS histology experiments should be described in the methods.

Figures:

15. Figures 2, 3, 4, and 5 and 6 for consistency (plus S2, S3, and S5): Graph coloring for Cre+ conditions need to be any other color besides green when the controls are labelled 'GFP'. It really creates a cognitively conflicting situation where the reader wants to think green represents green fluorescent protein (i.e. control), but it is the opposite. Switching green for grey, or using another color entirely, would really help.

16. Fig 1. Add a scale bar to magnified imaged in Quantification of smFISH. Oxtr foci per region using particle analysis should be done (as stated above).

17. Fig 2. [and others]: It is unclear from the behavioral schematic whether the object, social and anxiety behavioral testing was done over 3 separate days. Add Day 11, 12, 13 if that was the case.

18. Fig 2. please explicitly detail why there are different animal numbers reported for fig 2a-c (7/4), Fig 2d-g (11/6) vs supplemental fig 2(7/4), which all deal with behavior for AAV into DG animals. It should be mentioned in the results and methods whether separate cohorts of mice were used. If different cohorts were used for different behaviors, it makes the schematic in Fig2a seem not applicable to all cohorts. Also, add 5 min inter-trial interval ITI to NOR and Social discrimination tasks.

19. Fig 4. I would like to see representative images of CA2/CA3a, CA3b and CA3c for AA and AB. This will also help clarify why only CA3c was quantified. Why CA3c ensembles [and not CA2/CA3a or CA3b] were affected despite intact Oxtr expression should be discussed.

20. Fig FS1. It would be helpful to also include the cartoon indicating where the images were taken from in all cases, similar to what is in S1 c and d.

Author's Point-by-Point Response to Reviewers' Comments for Nature Communications submission

"Hippocampal oxytocin receptors are necessary for discrimination of social stimuli"
Raam et al.

We thank the reviewers for their invaluable and constructive criticisms. Extensively addressing all the concerns has made this manuscript significantly stronger and enabled us to give clarity and detail to anatomical data, as well as confirm the validity of our experimental manipulations. Based on our findings, we hope you will agree that the revised manuscript conceptually advances our understanding of the role(s) of hippocampal Oxytocin receptors in social memory processing.

In advance of conveying our rebuttal, we would like to note that in response to Reviewer #3's third point about the anatomical distinction of dorsal vs ventral, we have now changed all mentions of dorsal to "anterior" and all mentions of ventral to "posterior". This is stated here in order so as to avoid any confusion.

Responses to individual referees' concerns are documented below in blue text. 14 Rebuttal only figures (R1-R14) are attached to the end of this document.

Reviewer #1

Major concerns

1-1. Previous studies have suggested an important role for presynaptically localized OTR's. Consistent with these findings, in figure 1 and S1, Raam et al show that in both the ventral and dorsal CA2/3, OTR mRNA is densely expressed in the pyramidal cell layer, but does not take on a distribution consistent with cell body localization (in contrast to the CA1). Since previous studies have demonstrated that presynaptic terminals are capable of protein synthesis, it is possible that OTR mRNAs localized to CA2/3 are not made by hippocampal pyramidal neurons, but are instead made by inputs to the CA2 (e.g. the hypothalamus, which contains oxytocin neurons themselves producing autoreceptors). Although it is perhaps beyond the scope of this paper to ask for EM level characterization to differentiate these possibilities, electrophysiological characterization of miniature excitatory postsynaptic frequency and amplitude, combined with anterograde tracing from the hypothalamus (or other input regions) could clarify these possibilities, which dramatically alter the interpretation of the results, especially the cFOS studies in Figure 4, which are an indirect measure of neuronal activity in response to social stimuli.

R1-1. We agree with the reviewer that it is plausible that some of the Oxtr transcripts may reflect presynaptically localized Oxtr mRNAs. **However, the crux of the reviewer's concern lies in whether our viral manipulations target presynaptic Oxtrs in addition to postsynaptic Oxtrs in CA2/CA3.** To directly address this concern, we examined whether AAV₉- CamKII α -Cre is transported retrogradely from site of infection i.e. CA2/CA3. Analysis of every single brain of mice used in these experiments unequivocally confirms that the virus is not retrogradely trafficked to any input regions, including the DG and PVN (**Figure S4**). In this supplementary

figure, we show images of whole-brain GFP immunolabeling for a single animal illustrating the restriction of GFP to aCA2/3. Further, we have also included images of the injection site and aCA2/3 inputs for several other mice as rebuttal-only **Figures R1-6**. We are happy to include this data as part of supplementary figures upon request. **Because viral expression of Cre is restricted to aCA2/3 cell bodies (and not trafficked retrogradely) and recombination of Oxt is seen in CA2/CA3 (Fig. S4d,e) (and that this is a nucleus specific event), it is reasonable to rule out the possibility that knockout of Oxts in presynaptic terminals originating from the PVN is responsible for the behavioral and cellular effects.** The additional question of whether *Oxt* mRNA is also expressed in presynaptic terminals of CA2/CA3 afferents, while interesting, is beyond the scope of this study.

1-2. Although the authors would like to claim that molecular ablation and optogenetics experiments effectively localize their effects to hippocampal CA2/3 pyramidal cells, for reasons that are unclear, they have carried out their experiments using the AAV9 serotype, which has previously been shown to be taken up by presynaptic terminals and retrogradely transported 13,14. Thus the effects reported in figures 3, 4, and 6, may in fact reflect ablation or silencing of presynaptic OTRs originating outside the hippocampus, particularly in light of the distribution pattern of OTR mRNA in Figures 1 and S1 (discussed above). Thus the conclusions the authors draw would be significantly strengthened if the authors could show that these effects were absent following Cre or halorhodopsin expression using anterograde viruses into the hypothalamus and other input regions to the CA2/3, or retrograde viruses (e.g. CAV2, RbV, or psuedotyped LV) into the CA2/3.

R1-2. We appreciate the reviewer's important concerns about retrograde trafficking of AAV9 viruses, which we have used both for aDG deletion (Figure 2), and aCA2/3 deletion (Figure 3-4). We note that in Cook-Snyder et al. (reference 14 identified by the reviewer), AAV9s are only retrogradely transported at high titer, and upon dilution of the virus to 1:10 and 1:100, retrograde expression of Cre-GFP becomes very sparse. **We have used a dilution of 1:500 for AAV9-hSyn-Cre-GFP in the aDG knockout experiment and a dilution of 1:50 for AAV-CamKII-Cre-GFP expression for aCA2/3 knockout experiment and we do not observe any retrograde expression of GFP in inputs to aCA2/3 (Fig S4, rebuttal-only Fig R1-6) or aDG (Fig S2, rebuttal-only Fig R7-11).** The use of low titer virus may underlie the lack of retrograde expression in our manipulations. We have included extensive low and high magnification images of injection sites, DG inputs (entorhinal cortex, PVN, SUM, Medial septum), and aCA2/3 inputs (DG, entorhinal cortex, PVN) as part of the supplementary figures and several additional brains in rebuttal-only figures for the reviewers to assess. Optogenetic experiments (**Fig 5,6**) were carried out with AAV5 and we also do not observe retrograde transport of virus (**Fig S8**).

Minor concerns

1-3. It is not clear in Figure 2 why the authors used two different promoters (e.g. hSyn or CamKII α) to drive expression of Cre-GFP versus control GFP?

R1-3. We initially attempted to obtain hilar expression of Cre-GFP using the AAV9- CamKII α virus used in Figures 3-4, however Cre-GFP expression of this virus was restricted to the granule cell layer, rather than the hilus, where the majority of Oxts are expressed (**rebuttal-only Figure R12**). As we intended to drive Cre expression in hilar neurons where Oxts are expressed, we

obtained the hSyn-Cre-GFP virus. Because the hSyn-Cre-GFP and CamKII α -GFP viruses resulted in similar patterns of expression, we felt it was a minor concern to obtain a control GFP virus expressed under the hSyn promoter.

1-4. The authors argue that habituation is impaired only following OTR knockdown in dCA2/3 (figure 3f) but not dDG (figure 2g), but in the second trial habituation looks significantly impaired for dDG knockdown, and the effect is much less impressive in dCA2/3.

R1-4. The language used to describe these effects directly reflects statistical significance obtained using two-way ANOVA, as documented in the statistics spreadsheet. The two-way ANOVA interaction value for aDG knockout (**Figure 2**) is a trend of .0899, and thus no multiple comparison for each trial can be performed. We have changed the language of the manuscript to indicate a trend towards a deficit in habituation, rather than stating that there is no effect. The two-way ANOVA interaction value for aCA2/3 knockout (**Figure 3**) is .0053, indicating a significant effect.

1-5. In figure 4, the authors make a point of differentiating between CA2, CA3b and CA3c, but do not describe the functional or anatomical basis for defining these subregions.

R1-5. We thank the reviewer for the opportunity to clarify the anatomical boundaries identified, which are also raised by reviewer 3. Several independent studies have indicated that the CA3 area is anatomically and functionally heterogeneous, with the area closest to the DG hilus (commonly referred to as CA3c, or “proximal CA3”) supporting pattern separation, and the region further from the DG (referred to as CA3ab, or “distal CA3”), contributing to pattern completion^{1,2,3}. Considering this functional heterogeneity, we chose to analyze both subregions of CA3 for the catFISH experiment in Figure 4. We appreciate that exact anatomical boundaries are not feasible to identify in the absence of antibodies specific to CA3 subregions (as pointed out by reviewer 3), and have thus changed the wording in our manuscript to “CA3_{proximal}” and “CA3_{distal}” to accommodate a more flexible definition of these subregions. We have also used this terminology in Figure 1, where we have added additional analysis of Otr mRNA expression in CA3 subregions. This terminology is consistent with other papers in the field, mentioned above, and we hope the reviewers will agree with this change in nomenclature.

1. Scharfman, “The CA3 ‘Backprojection’ to the Dentate Gyrus.” *Prog Brain Res*, 2007.
2. Lee *et al.*, “Neuronal Population Evidence of Functional Heterogeneity Along the CA3 Transverse Axis: Pattern Completion vs Pattern Separation.” *Neuron*, 2015.
3. Sun *et al.*, “Proximodistal Heterogeneity of Hippocampal CA3 Pyramidal Neuron Intrinsic Properties, Connectivity, and Reactivation during Recall.” *Neuron*, 2017.

Reviewer #2

2-1. In Figure 1, the authors showed an overlap of OTR expression with Gad1 both within dorsal and ventral CA2/3. The presence of even a minor population of inhibitory cells receptive to OT in this region might have indeed an important role not only in intra-hippocampal circuitry, but also in hippocampal output structures. Taking into account the growing evidence showing unique properties of CA2 interneurons, it would be reasonable to show the type of

interneurons expressing OTR in the CA2/CA3 region. A feasible solution of this question, in my opinion, is to employ staining of calcium-binding proteins to identify subclasses of local interneurons expressing OTR.

R2-1. We agree that this is a valuable question for the field, and as requested of us, **we have carried out multiplex FISH for *Oxtr*, *Parvalbumin*, and *Somatostatin* to characterize the subclasses of interneurons expressing *Oxtr*. Additionally and importantly, we have carried out colocalization analysis for these markers in DG, CA2/3, and CA1 in both anterior and posterior hippocampus in age matched male and females mice.** All analyzed regions of hippocampus showed *Oxtr*⁺ cells as a mixed population of PV⁺, SST⁺, and PV⁻SST⁻ cells. Analysis of *Oxtr*, PV and SST expression in aDG showed greater colocalization of *Oxtr* and SST than *Oxtr* and PV (Fig 1c,d). Further, a large percentage of *Oxtr* cells were negative for both PV and SST, indicating the presence of additional subclasses of *Oxtr*⁺ interneurons that are not captured in PV and SST populations. Characterization of *Oxtr*⁺ cells in pDG revealed a mixed population of PV⁺ and SST⁺ cells, with a larger fraction of *Oxtr*⁺ cells expressing SST than PV (Fig S1c,d). Assessment of overlap between *Oxtr*, PV and SST in anterior and posterior CA2/CA3^{distal} revealed a larger fraction of PV⁺ *Oxtr*⁺ cells than SST⁺*Oxtr*⁺ cells (Fig 1i,j, S1g,h).

In addition, we have carried out colocalization analysis for *Oxtr/Gad1* and *Oxtr/PV/SST* in age-matched female mice to determine sexual dimorphism in *Oxtr* expression. We do not observe sexual dimorphism in *Oxtr* expression in the anterior and posterior hippocampus in any subfield (consistent with results obtained for *Oxtr* distribution in CA3 assessed by radioligand binding in Hammock et al, *Frontiers Beh Nsc* 2013). All analyzed regions of anterior and posterior hippocampus showed similar levels of *Oxtr* mRNA expression between males and females, and colocalization analysis also revealed no difference in identity of cells as excitatory (*Gad1*⁻), inhibitory (*Gad1*⁺), or positive for Parvalbumin or Somatostatin. These data are collectively presented in **Figures 1 and S1**.

2-2. The authors claim that the OTR expression in dCA1 is absent. Since the effects of TGOT on hippocampal oscillatory activity in CA1 were described in the literature, and previous in situ studies indeed showed the presence of OTR mRNA in the principal cell layer of CA1, it would be important to demonstrate OTR mRNA expression in mouse brain planes containing both dorsal and ventral CA2/3 (Br level: -5; -5,7).

R2-2. The reviewer seems to be referring to studies carried out in rat brain tissue (eg: Tribollet et al, *Brain research*, 1988), in which *Oxtr* expression is observed in aCA1, while our studies (and all the knock-in studies) are carried out in mouse brain tissue (also see receptor binding studies to directly compare with rat studies, eg: Hammock et al, *Frontiers Beh Nsc*, 2013). For the referee, we have included quantification of mean *Oxtr* intensity, corrected for background, in Fig 1f and Fig S1j. Representative images are also included. These figures quantitatively demonstrate enrichment of *Oxtr* mRNA in pCA1v, but not aCA1 or pCA1d.

2-3. In the experiments showing impairment of social stimuli discrimination after acute blockade of OTR in CA2/3, the authors used OTR antagonist L-368, 899 Hydrochloride. How did the authors determine the spread of OTR-A? It is difficult to imagine that infusion of OTR-A will affect only the dCA2/CA3 as the compound is characterized by great potency and extremely rapid brain penetration.

R2-3. We appreciate the reviewer's concern for rapid brain penetration with the Oxt receptor antagonist L-368, 899 Hydrochloride. Other oxytocin pharmacological studies have used the same antagonist at the same concentration and infusion protocol^{1,2}. We agree that spread of a drug is generally a concern with pharmacological infusion, but maintain that this experiment is an important one to corroborate the results from genetic deletion of *Oxtr* in **Figure 3**. As such, we have included it in the supplementary figures rather than the main figures so as not to overstate its significance.

1. Dolen *et al.*, Social reward requires coordinated activity of nucleus accumbens oxytocin and serotonin. *Nature*, 2013.
2. Najakima *et al.*, Oxytocin modulates female sociosexual behavior through a specific class of prefrontal cortical interneurons. *Cell*, 2014.

2-4. The authors should confirm that the rAAV9, equipped with CamKII, is exclusively expressed in principal cells. From my experience, infection with CamKII-rAAV1/2 leads to expression of GFP also in interneurons.

R2-4. We have used multiplex FISH for *Cre* and *Gad1* mRNA to determine whether or not the CamKII α -Cre-GFP virus also infects interneurons. We do observe minimal colocalization of *Cre* and *Gad1* in CA2/3, as the reviewer has noted. Cells that were double-positive for *Cre* and *Gad1* made up 7% of all total *Cre*⁺ cells. This overlap is minimal, but not insignificant, and we have thus included these data as part of **Figure S5b** as well as in the main text. We have changed our wording for Figures 3 and 4 to indicate that *Oxtr* recombination was achieved in mostly excitatory and a small number of inhibitory cells of the principal cell layer of CA2/3, rather than stating exclusive principal cell infection. Thank you for raising this concern as this information is useful to the larger neuroscience community.

2-5. To confirm a critical role of the OTR dCA2/3 – dCA1 circuit in the modulation of social discrimination, the gain of function experiment (ChR2) should be added as an essential control.

R2-5. We appreciate the suggestion but think (based on our own experience and not for a lack of effort!) that ChR2 dependent stimulation of terminals will not permit straightforward interpretation of the data.

- a) It has been well documented that ChR2 driven terminal stimulation can lead to back-propagating axon potentials that may activate cell bodies (and projections to other brain regions), therefore defeating the terminal specificity desired by such a manipulation^{1,2}. Unpublished data from our lab has confirmed this. **When stimulating axon terminals from aCA2/3 to the dorsolateral septum (DLS), we observe robust hyperactivation of the DG along the anterior-posterior axis. These data are summed up in rebuttal-only Figure R13.**
- b) Furthermore, optogenetic studies have indicated that while NpHR silencing of hippocampal circuits can serve as effective loss-of-function manipulations, ChR2 manipulations, rather than enhancing hippocampal function, often impair it³. This may be due to difficulty reproducing endogenous patterns of activity with ChR2, especially critical when stimulating terminals. Finally, we do not think that pharmacological silencing of cell bodies while stimulating terminals remedies these issues. For the

reasons summarized here and our data, we think that Chr2 stimulation of aCA2/3 terminals to pCA1 would not edify our knowledge of how the circuit behaves endogenously during behavior.

1. Grossman *et al.* The spatial pattern of light determines the kinetics and modulates backpropagation of optogenetic action potentials. *J Comput Neurosci*, 2013.
2. Li *et al.*, Fig S5. “A motor cortex circuit for motor planning and movement” *Nature*, 2015.
3. Kheirbek *et al.*, “Differential control of learning and anxiety along the dorso-ventral axis of the dentate gyrus.” *Neuron*, 2013.

Reviewer #3

3-1. Throughout, the authors refer to the area of densest *Oxtr* expression as area CA2/3, yet the authors appear to also be aware that there are huge differences between high *Oxtr* expression levels in CA3a (near CA2), and CA3b and c, which are quite a bit lower. Therefore it would be appropriate that the authors distinguish CA3a and use a more accurate description throughout: CA2/CA3a (vs. CA2/CA3). Second, it would also seem worthwhile to address why CA3b, with its intermediate levels of *Oxtr* (at best), is grouped with CA3a, and not CA3c, for the catFISH experiments in Fig 4, even if was simply for technical reasons relating to the injections.

R3-1. We thank the reviewer for the opportunity to clarify the nomenclature used in the manuscript. For reasons noted above in response to comment 1-5, we have chosen to use the terms “CA3_{proximal}” and “CA3_{distal}” instead of CA3c and CA3ab throughout the manuscript. We have accordingly grouped CA3_{distal} with CA2 and collectively called this region “aCA2/CA3_{distal}” due to the similar levels of *Oxtr* expression in both regions, and the fact that our injections in Figures 3, 5, and 6 target both CA2 and CA3_{distal}. The reviewer is correct that CA3_{proximal} has notably lower *Oxtr* expression (quantified in **Figure 1f**), and we have therefore distinguished the two regions from each other throughout the manuscript.

3-2. In a related topic, the authors are using RNAscope, so they should be able to accurately quantify the density of transcripts in each region for figure 1: CA1, CA2/CA3a, and CA3c (ideally also CA3b separately, but this would require some markers or clear anatomical features distinguishing the regions). Also, for figure 1b and c, similar pie charts as in a).

R3-2. We have revised Figure 1 to include mean *Oxtr* intensity (corrected for background) for CA1, CA2/3_{distal}, and CA3_{proximal}. Schematics illustrating the subregions is included in **Fig 1l**. *Oxtr* is notably enriched in CA2/3_{distal}, while CA1 and CA3_{proximal} yielded intensity levels comparable to background. We have also quantified *Oxtr/Gad1* overlap for aCA2/CA3_{proximal} and shown pie charts, indicating that roughly 10% of the *Oxtr* expressing cells are *Gad* positive. Such a quantification is not meaningful for aCA1, as *Oxtr* mRNA levels are not any higher than background. These data are summarized in **Figure 1**. Similar quantifications have been carried out for posterior hippocampus in **Figure S1**.

3-3. The authors refer to a vCA1d, which is really just dorsal CA1. Just because it is further back in a coronal section does not make it in any way ‘ventral’. At best, it is mid-way along the hippocampal axis, but at that (horizontal) level, the dentate is clearly a ‘V’ shape (vs. a ‘U’ seen in the ventral hippocampus). If the authors have horizontal sections showing dCA2/CA3a

projections in vCA1, they would also be very useful to see (perhaps in Fig 5b). Again- some quantification of the Oxtr expression in ventral CA1 would be required if the authors are to claim Oxtr are higher in the ventral CA1. Because determining subfields in ventral hippocampus is very difficult in coronal sections, the authors would ideally quantify Oxtr levels in horizontal sections. At least, though, they should use an adjacent section stained for a CA1 marker such as wfs1 to confirm what they are looking at is actually CA1. It looks like they could possibly be looking at the subiculum, depending on the sections.

R3-3. We stand corrected that “vCA1d” is still dorsal within the section, and have therefore changed our nomenclature to anterior vs posterior hippocampus. We have shown more detailed images of the projection from aCA2/3_{distal} to pCA1d (**Figure S8**), including the direct overlap of this projection with CA1 marker WFS1 in both coronal and horizontal planes (**Figure S10**). Using the nomenclature pCA1d and pCA1v should therefore be more accurate to distinguish between the more dorsal area of pCA1 which is innervated by aCA2/3_{distal}, and the more ventral region, which is not. We did not quantify Oxtr mRNA in horizontal tissue for the RNAscope experiments, however we did confirm that we were imaging CA1 and not subiculum by using tissue stained for WFS1 as a boundary indicator.

3-4. The authors focus their viral recombination/Oxtr deletion injections in either dentate gyrus or dorsal CA2/3a, but because they are not genetically targeted, the authors need to provide more images, including a panel of lower magnification images for the readers to get a better understanding of where exactly those injections were landing, and whether the cre is specifically targeted. Drawings, such as in Fig S1b and c, would also be helpful in many cases. Specifically, with regard to the dCA2/3a injection, is Figure 3b representative (it appears to largely be in CA2)? Figure 5b, on the other hand, would be improved by showing multiple planes of sections at low power, and indicating the injection site and projections (in higher power). In addition, there is some concern that the DG injections were not specific, given a peculiar affinity CA2 neurons have for AAVs. Without genetically targeted Cre expression, the documentation of the injections should be more rigorous and inclusive. The additional images would be appropriate for supplementary material if not the main figures.

R3-4. We agree that additional images are necessary to assist the reader in understanding the specificity of our manipulations, and we have provided low and high magnification images of the injection sites for both the DG and CA2/3 deletion experiments in **Fig S2** and **Fig S4**, and optogenetic terminal silencing in **Fig S8**. Schematics indicating the regions of interest have been provided in **Fig 1** and can be referenced throughout the paper.

With regards to the aCA2/3_{distal} manipulation, the reviewer is correct that the CamKII α - Cre injections encompass the region labeled by CA2 marker RGS14 and about 10% of the Cre-expressing cells are located in CA3. The lack of a CA2-specific mouse line for this study requires us to be more general in our claims, rather than exclusively claiming CA2. For this reason we find it more accurate to refer to both CA2 and CA3.

With regards to the aDG manipulation, we have demonstrated in **Fig S2** and **rebuttal-only Figures R7-11** that we do not observe Cre-GFP expression in CA2 and our injections are specific to aDG.

3-5. For the optogenetic silencing experiment in the dCA1 projection target from dCA2/3a

(Figure 5) the authors don't show cFos data to confirm silencing (which they did do in the DLS and vCA1 experiments). Why not? The silencing of CA2/CA3a may very well increase fos, given one idea in the field that CA2 output has a net inhibitory effect on at least some neurons in CA1, but the reasons for why this data was excluded should at least be given.

R3-5. For reasons that are unclear to us, we did not observe any effect on cFos when silencing the projection from aCA2/3_{distal} to aCA1. This may be because of the time interval (post-illumination and pre-perfusion) that was used. However, we did observe silencing for the projections to pCA1 (**Fig 6**) and DLS (**Fig S9**), using the same virus, titer, and injection protocol. This is likely due to differences in cFos temporal dynamics from region to region. A 60 minute protocol was used for all three experiments, however it is possible that aCA1 may have slower or faster temporal dynamics than the other regions tested, or that a more sensitive measure, such as electrophysiological recordings, may be necessary to capture an effect on silencing in this region. Because the role of CA1 in object memory has been well-established^{1,2,3,4}, the behavioral phenotype observed is expected and consistent with the literature, and we feel that a lack of an effect on cFos in this particular does not invalidate the finding.

1. Cohen *et al.*, "The Rodent Hippocampus is Essential for Nonspatial Object Memory", *Current Biology*, 2015.
2. Cohen and Stackman. "Assessing rodent hippocampal involvement in the novel object recognition task." *Behavioural Brain Research*, 2015.
3. Warburton and Brown. "Neural circuitry for rat recognition memory." *Behavioural Brain Research*, 2015.
4. Broadbent *et al.*, "Object Recognition and the Rodent Hippocampus." *Learning and Memory*, 2010.

3-6. Ideally, if the authors still have the sections from the deletion experiments, they should confirm actual knockdown, either with RNAscope or with an antibody (if they can acquire one, from Mitre, *et al.*, for example).

R3-6. The Froemke lab generously provided us with aliquot of their *Oxtr* antibody. However, we were unable to obtain specific labeling as shown in **rebuttal-only Fig R14**. Specifically, we see labeling that is inconsistent with *Oxtr* mRNA expression (as evident in labeling of DG granule cell layer, aCA3proximal, and aCA1, three areas in which *Oxtr* mRNA is nearly absent in mice) documented in Fig 1 and other published studies using *Oxtr* Knock-in reporter lines. This may be due to variability in specificity among different batches of antibody, and therefore, we do not think that this particular batch of antibody in our hands is useful for validation of *Oxtr* recombination.

Instead, we have utilized RNAscope to quantify *Oxtr* expression in *Oxtr*^{+/+} vs. *Oxtr*^{f/f} mice injected with Cre virus into aCA2/CA3_{distal} and aDG. In order to determine viral mediated recombination of *Oxtrs*, we quantified the percentage of DAPI+ cells expressing *Oxtr* mRNA in aDG hilar neurons of *Oxtr*^{+/+} vs. *Oxtr*^{f/f} mice injected with AAV₉-hSyn-Cre. *Oxtr*⁺ cells comprised 11% of all DAPI+ cells in *Oxtr*^{+/+} mice and 6% of all DAPI+ cells in *Oxtr*^{f/f} mice (**Fig S4d,e**) suggestive of *Oxtr* recombination (**Fig S2f,g**). To determine recombination of *Oxtrs* in aCA2/CA3_{distal}, we quantified the percentage of DAPI+ cells expressing *Oxtr* mRNA in aCA2/CA3_{distal} *Oxtr*^{+/+} vs. *Oxtr*^{f/f} mice injected with AAV₉-CamKII α -Cre. *Oxtr*⁺ cells comprised

50% of all DAPI+ cells in *Oxtr*^{+/+} mice, and 25% of all DAPI+ cells in *Oxtr*^{f/f} mice (Fig S4d,e). Interestingly, while we observed fewer total DAPI+ cells that were positive for *Oxtr* in *Oxtr*^{f/f} mice, we observed larger *Oxtr* puncta in cells that retained *Oxtr* mRNA suggestive of redistribution of transcripts.

Minor:

3-7. Please be consistent in using the correct nomenclature for rodent genes (italics, first letter of symbol in caps) vs proteins (not italics, all caps).

R3-7. Thank you. We have changed all gene and protein names to the appropriate nomenclature.

3-8. Pg 3, first paragraph second sentence: this is not the definition of global remapping. Two non-overlapping, sparsely encoded ensembles is the proposed mechanism behind pattern separation. Global remapping refers to modifying an existing ensemble [by rate or field changes] to reflect updated context. See Colgin et al 2008 TINs review currently cited as #48.

R3-8. Duly noted. We have removed the reference to global remapping in this sentence and elsewhere.

3-9. same paragraph- It is important to distinguish when the species differs in the studies used for the rationale.

R3-9. We have added references to species used for the references related to social memory and oxytocin signaling.

3-10. Last sentence pg 5: Object discrimination was not tested with pharmacology.

R3-10. We have changed the wording in this sentence to clarify the intended meaning, and have reframed it as a standalone sentence that summarizes the whole section on aCA2/3_{distal}, rather than the previous format which could have been misunderstood as only referring to the pharmacology experiment.

3-11. Pg 6, “well-characterized Schaffer collateral pathway from dCA2/CA3-dCA1”. As this projection populates primarily the stratum Oriens in CA1, I don’t think the dCA2->CA1 projection is widely considered to be a part of the Schaffer collaterals, and anything from CA2 is far from being ‘canonical’. Therefore I suggest the authors use a less controversial description such as, “we first targeted the projection from dCA2/CA3a to dCA1.”

R3-11. We agree that this description is more accurate and conservative and have changed our wording to reflect this.

3-12. In general, the authors should be wary of how they use of the term “retrieval” in describing the results of the novel object recognition experiments. Given that the time point for testing is 30 min post first exposure/encoding, the authors are not testing retrieval in the way it is more typically performed (i.e. more usual is 24 hours post-encoding to test long term memory retrieval). Because of this different methodology, the reader is unable to extrapolate the results to the large amount of literature on this subject. Furthermore, the possible plasticity mechanisms/brain regions involved at this time point are not clear or not discussed. The

authors should consider discussing this distinction in the context of OXTR signaling as well as their reasoning for choosing this time point.

R3-12. We have only used the word “retrieval” in reference to the social discrimination task and not the NOR task. The reasoning for choosing a 30 minute time point for the novel object task was primarily for the purpose of maintaining consistency with the time point used for social the discrimination task. Had a 24 hour time point been used, it would be difficult to make any meaningful conclusions about how the circuits are behaving relative to social vs non-social modalities, rather than simply short-term vs long-term memory. We have updated the main text to include the rationale for choosing this time point. We understand that the object recognition literature is a vast and nuanced one, however because the role of CA1 in this task is well described in the literature (**references cited in comment 3-5**) and corroborated by our optogenetic findings in **Figure 5**, we feel an in-depth discussion of circuits underlying short-term object memory is tangential to the main findings.

3-13. It would be important to discuss why CA2 was not assessed in the catFISH experiments. This is likely due to the fact that cFos foci are very small in CA2 and so are difficult to quantify (Alexander, et al.), but this should be explicitly stated given that Oxts were knocked out of CA2/CA3a and no affect was found for the CA3ab region. Furthermore, it is unclear why there would be an effect in CA3c. This should also be discussed.

R3-13. You are correct. We intended to quantify c-Fos foci in CA2 however too few foci were observed. An average of only 2 cFos+ foci/section renders the data too noisy (with the given numbers of mice used here) to be meaningfully quantified. As noted in **R1-5** above, the functional heterogeneity of CA3 documented by several labs notes CA3_{proximal} (noted as CA3c in the first version of the manuscript) as a robust pattern separator, at both the electrophysiological and behavioral level, while CA3ab, which consists of strong recurrent collaterals, functions more as a pattern completion circuit. Our data is consistent with this finding and suggests that CA3ab does not segregate distinct social stimuli at the cellular level, even in control animals. Rather, our data suggests that CA3c (CA3_{proximal}), due to its intrinsic firing properties that are similar to the DG, is the locus for population-based encoding of social stimuli in controls, and this region is where we observe a disruption in cellular reactivation in response to social stimulus re-exposure (AA group) when Oxts are deleted. That CA3_{proximal} is the locus in which we observe an encoding phenotype despite not being the locus of Oxt expression suggests a role for local microcircuits projecting from CA2/3_{distal} to CA3_{proximal}, such as through local interneurons that share reciprocal connections with CA3 pyramidal neurons, both receiving pyramidal input and projecting back onto pyramidal neurons. We have updated the discussion in the main text to more clearly reflect these ideas.

3-14. There is no mention in the discussion of the fact that attenuation of dCA2/3 axons in vCA1 (using the dCA2/3 eNpHR3) showed no significant difference in the social discrimination test itself (Fig 6g). Is the discrimination ratio difference enough to state the importance? If the eNpHR positive fibers only represent a fraction of the dCA2/3 to vCA1 fibers that produced the robust effects in the Oxt deletion experiment (Fig. 2g), that should be addressed.

R3-14. Discrimination ratio is generally a more stringent test as it takes into account each animals behavior with familiar and novel stimuli. Nevertheless, independent of the

discrimination ratio, the 2way-ANOVA interaction is significant for this behavioral test ($p=.0033$), indicating that the two groups are behaving differently. We agree with you that the modest discrimination seen in the experimental group may reflect insufficient coverage of the large termination zone in pCA1d, the high likelihood that silencing is not complete and that we are not targeting just the *Oxtr* expressing neurons in aCA2/3_{distal}. This observation is now noted in the discussion.

3-15. How does the retrieval phase of the social discrimination task (p5, during pharm block of OXTRs) compare with the test phase of the task (p7, during circuit attenuation)? I'm assuming these are the same periods of the same task.

R3-15. Yes, these are the same periods of the same task. "Retrieval" is used in the context of the pharmacology experiment in order to emphasize the distinction between acquisition and retrieval.

3-16. The authors should discuss their findings in the context of a recent oxytocin receptor paper Ripamonti, et al. 2017 and/or the sexual dimorphism seen in *Oxtr* distribution (Mitre, 2016)

R3-16. We have cited the Ripamonti et al. study in the discussion section.

Methods:

3-17. Were multiplex bacterial negative controls performed alongside the smFISH experiments to appropriately control for background during image acquisition? If so, describe.

R3-17. Bacterial negative controls were run alongside initial piloting of the RNAscope assay to ensure that background was comparable to negative controls, however an additional negative control slide was not run alongside each additional assay for *Oxtr/Gad1* and *Oxtr/PV/SST*. Variation in background was minimal, but was controlled for by background subtraction in *Oxtr* intensity quantification. Additional controls were run of each probe individually to ensure no overlap across probe channels. We also ran each probe in different fluorescent channels to ensure that results were comparable across fluorophores.

3-18. For catFISH experiments, given the differences in cell number due to difference in section/angle and CA3ab vs CA3c, it would be preferable to present the data as % of cells instead of (or in addition to) # of cells per section. At the very least, provide the dimensions of the ROI assessed (i.e. 500microns squared) and provide an average number of cells assessed per region so that percentages can be inferred by a knowledgeable reader.

R3-18. We have quantified total number of DAPI positive cells in the ROI for proximal and distal CA3 and have included this in the Methods section.

3-19. For DG FOS histology experiments, please elaborate whether both blades were counted/assessed.

R3-19. Yes, both blades were assessed. Methods have been updated to indicate this.

3-20. Fig 2. The time-line for FOS histology experiments should be described in the methods.

R3-20. We have updated the methods to include the timeline for cFos histology experiments.

Figures:

3-21. Figures 2, 3, 4, and 5 and 6 for consistency (plus S2, S3, and S5): Graph coloring for Cre+ conditions need to be any other color besides green when the controls are labelled 'GFP'. It really creates a cognitively conflicting situation where the reader wants to think green represents green fluorescent protein (i.e. control), but it is the opposite. Switching green for grey, or using another color entirely, would really help.

R3-21. We respectfully choose to keep the color scheme as is. Both the control and experimental groups are injected with a green GFP virus. We chose to represent the control group as grey and the experimental group as green in order to represent the absence vs the presence of a manipulation. In order to avoid potential confusion on the reader's part, we have changed all color legends in each figure to denote "GFP" and "Cre-GFP" instead of simply "GFP" and "Cre". This color scheme is consistent throughout the manuscript and we hope it will be straightforward to follow with this adjusted labeling.

3-22. Fig 1. Add a scale bar to magnified imaged in Quantification of smFISH. Oxtf foci per region using particle analysis should be done (as stated above).

R3-22. We have added scale bars to magnified images and have performed intensity analysis to quantify *Oxtf* density per region.

3-23. Fig 2. [and others]: It is unclear from the behavioral schematic whether the object, social and anxiety behavioral testing was done over 3 separate days. Add Day 11, 12, 13 if that was the case.

R3-23. We have changed all behavioral figures and supplementary figures to denote exact days in which behavioral tests were performed.

3-24. Fig 2. please explicitly detail why there are different animal numbers reported for fig 2a-c (7/4), Fig 2d-g (11/6) vs supplemental fig 2(7/4), which all deal with behavior for AAV into DG animals. It should be mentioned in the results and methods whether separate cohorts of mice were used. If different cohorts were used for different behaviors, it makes the schematic in Fig2a seem not applicable to all cohorts. Also, add 5 min inter-trial interval ITI to NOR and Social discrimination tasks.

R3-24. We used two separate cohorts for Figure 2, and anxiety related behaviors (OFT and EPM) were only carried out for the first cohort. We therefore have removed any mention of anxiety on the main Figure 2 since it does not apply to all the animals in that figure. We make mention of anxiety only in the supplemental figure (which is now **Fig S3**). Fig 2a-c, the cFos experiment, also only included one of the two cohorts, however we have only included the cFos timeline directly next to that panel rather than in panel a, so that it does not seem to imply that all animals underwent the cFos protocol. **We have now ensured that all experimental timelines in the top of each figure apply to every single animal in that figure and have noted in the methods why Ns are variable for the DG knockout experiment.** We have also added the 5-min ITI to all figures.

3-25. Fig 4. I would like to see representative images of CA2/CA3a, CA3b and CA3c for AA and

AB. This will also help clarify why only CA3c was quantified. Why CA3c ensembles [and not CA2/CA3a or CA3b] were affected despite intact Oxtr expression should be discussed.

R3-25. We have added representative images of catFISH sections in **Figure S7**. For clarity, CA3c and CA3ab were both quantified, however a significant effect was only observed in CA3c (now CA3proximal). All of the data showing a lack of effect in CA3ab (now CA3distal) is shown in Figure 4. We have elaborated on potential mechanisms underlying an effect on CA3c rather than CA3ab (where Oxtr expression is densest) in the discussion, and above in **R3-13**.

3-26. Fig FS1. It would be helpful to also include the cartoon indicating where the images were taken from in all cases, similar to what is in S1 c and d.

R3-26. We have included color-coded schematics of ROIs within anterior and posterior hippocampus **Figure 1L**.

Figure R1

Figure R2

Figure R3

Figure R4

Figure R5

Figure R6

Fig R7

Figure R8

Figure R9

Figure R10

Figure R11

Figure R12

30 min. cFos - ChR2 stimulation of aCA3 to Dorsolateral Septum (DLS)

S: Sham
Y: EYFP
C: ChR2

Figure R14

Oxtr antibody

REVIEWERS' COMMENTS:

Reviewer #2 (Remarks to the Author):

The authors fully addressed my concerns.

Reviewer #3 (Remarks to the Author):

The authors have adequately address my concerns, and in my opinion the concerns of the other reviewers. The study provides an important contribution to the literature on the relationship between hippocampal OxtRs, dorsal/ventral hippocampal circuitry, and social behavior.

I only have one remaining minor issue:

page 7, first paragraph: "Interestingly, in the elevated plus maze, we observed a decrease in time spent in the closed arm..." Did the authors intend 'increase' here?

Reviewer #3 (Remarks to the Author):

The authors have adequately address my concerns, and in my opinion the concerns of the other reviewers. The study provides an important contribution to the literature on the relationship between hippocampal OxtRs, dorsal/ventral hippocampal circuitry, and social behavior.

I only have one remaining minor issue:

page 7, first paragraph: "Interestingly, in the elevated plus maze, we observed a decrease in time spent in the closed arm..." Did the authors intend 'increase' here?

Author response:

Thank you for pointing this out, you are correct, and we have edited the wording to correctly reflect the EPM data in Fig S5.